# What's the Harm? Sharp Bounds on the Fraction Negatively Affected by Treatment

**Nathan Kallus**
Netflix and Cornell University
`kallus@cornell.edu`

## Abstract

The fundamental problem of causal inference – that we never observe counterfactuals – prevents us from identifying how many might be negatively affected by a proposed intervention. If, in an A/B test, half of users click (or buy, or watch, or renew, *etc.*), whether exposed to the standard experience A or a new one B, hypothetically it could be because the change affects no one, because the change positively affects half the user population to go from no-click to click while negatively affecting the other half, or something in between. While unknowable, this impact is clearly of material importance to the decision to implement a change or not, whether due to fairness, long-term, systemic, or operational considerations. We therefore derive the *tightest-possible* (*i.e.*, sharp) bounds on the fraction negatively affected (and other related estimands) given data with only factual observations, whether experimental or observational. Naturally, the more we can stratify individuals by observable covariates, the tighter the sharp bounds. Since these bounds involve unknown functions that must be learned from data, we develop a robust inference algorithm that is efficient almost regardless of how and how fast these functions are learned, remains consistent when some are mislearned, and still gives valid conservative bounds when most are mislearned. Our methodology altogether therefore strongly supports credible conclusions: it avoids spuriously point-identifying this unknowable impact, focusing on the best bounds instead, and it permits exceedingly robust inference on these. We demonstrate our method in simulation studies and in a case study of career counseling for the unemployed.

## 1   Introduction

Before making changes to an online platform, product managers regularly conduct experiments ("A/B tests") to make sure a change does not negatively impact users' experience. Similarly, extensive program evaluations, whether experimental or observational, are a corner stone of evidence-based policymaking and help avoid harmful policy changes. These tests and evaluations focus on assessing the causal effect of a change on key metrics, whether user engagement or retention on online platforms or outcomes like employment for social programs. Average treatment effects, heterogeneous treatment effects, and distributional treatment effects on these outcomes quantify the quality of the intervention in terms of aggregate impacts on the population or on subpopulations. This is in its essence a statistical solution to the *fundamental problem of causal inference*: assess aggregate effects since the impact on any one individual can never be observed. We can never know whether a user that churned under arm "B" would have been retained under arm "A," but we can characterize the distribution of retention under each arm, even segmenting the population by observable features.

While unknowable, the impact of changes on individuals is clearly of material importance to the decision to implement a change or not. This individual impact can have downstream effects, whether user-behavioral, reputational, or operational. Moreover, quantifying how negative this impact can be is crucial to understanding fairness with respect to the welfare of the individuals, rather than just

36th Conference on Neural Information Processing Systems (NeurIPS 2022).

focusing on social (*i.e.*, average) welfare, the welfare of the decision-making entity, or the welfare of observable groups. Specifically, even if the average treatment effect is zero, there can still be a sizable subpopulation that is negatively affected by the change. That is, a change with zero average effect cannot in earnest be called "harmless." In this paper we consider measuring the fraction that are negatively affected. The aim is to be able to flag changes that may individually harm many. Unfortunately this fraction cannot be identified from data, even experimental data, but we may still be able to bound it. For example, if, in an A/B test, half of users click (or buy, or watch, or renew, etc.) whether exposed to the standard experience "A" or a new one "B," hypothetically it could be because the change affects no one, because the change positively affects half the user population to go from no-click to click while negatively affecting the other half, or anything in between. Such a large bound on the range of possibilities, however, is very crude and uninformative.

In this paper, we derive the *sharp* bounds on the fraction of individuals negatively affected by treatment using all baseline features available, and we develop tools to conduct robust inferences on these bounds. That is, we characterize the *tightest-possible* interval that contains all values of this unknowable quantity consistent with all observable information, in particular crucially leveraging unit feature information. We consider both the fraction negatively affected under a wholesale change from one treatment regime to another as well as under an optimal policy that assigns each observable subpopulation the treatment with best average outcomes for the subpopulation. Then, we tackle how to actually assess these theoretical bounds based on actual data, namely, how to estimate the interval endpoints and construct confidence intervals (CIs) that account for sampling error, on top of the inherent epistemic uncertainty characterized by the bounds. The bounds involve potentially complex functions such as the conditional average treatment effect (CATE) function, so we develop inference methodology that is exceedingly robust to learning these functions. Namely, our inferences are valid and calibrated even when these complex functions are learned nonparametrically at slow rates, and our estimated bounds remain valid, albeit conservative, even when these are learned inconsistently, so that conclusions as to the presence of harm based on our tools can be highly credible. We demonstrate our tools in a simulation study and in a case study of French reemployment assistance programs.

## 2  Problem Set Up and the Fraction Negatively Affected

**The Population**    Each individual in the population is associated with baseline **covariates** (observable characteristics), $X \in \mathcal{X}$, and two *binary* **potential outcomes**, $Y^*(0), Y^*(1) \in \{0, 1\}$, corresponding to each of two treatment options, 0 and 1. Examples include: two versions of product on an online platform and whether user is retained in the next quarter, or two versions of a job training program for the unemployed and whether the participant is reemployed within the following year. We will consider the generalization to non-binary outcomes in Remark 1 and Thm. 3.

The **individual treatment effect** is the difference in potential outcomes: ITE $= Y^*(1) - Y^*(0)$. We assume that an outcome value of 1 corresponds to a better outcome (*e.g.*, retained, reemployed) and 0 to a worse outcome (*e.g.*, churned, unemployed). Thus, if treatment 0 represents status quo and treatment 1 represents a proposed change, then individuals with ITE $= 0$ ($Y^*(0) = Y^*(1) = 0$ or $Y^*(0) = Y^*(1) = 1$) are unaffected by the change, individuals with ITE $= 1$ ($Y^*(0) = 0$, $Y^*(1) = 1$) benefit from the change, and individuals with ITE $= -1$ ($Y^*(0) = 1$, $Y^*(1) = 0$) are harmed by the change. This list of potential-outcome combinations is exhaustive.

**The Data**    We consider data from either a randomized experiment or an observational study, wherein we never observe both $Y^*(0)$ and $Y^*(1)$ simultaneously. Each individual is associated with a **treatment** $A \in \{0, 1\}$, and we observe the **factual outcome** $Y = Y^*(A)$ corresponding to $A$. We never observe the counterfactual $Y^*(1 - A)$. That is, the data consists of observations of $(X, A, Y)$.

In particular, the distribution of the data only reflects part of the full population distribution involving both potential outcomes. Let $\mathbb{P}^*$ denote the distribution of $(X, A, Y^*(0), Y^*(1))$. Define the coarsening function $\mathcal{C} : (x, a, y_0, y_1) \mapsto (x, a, ay_1 + (1 - a)y_0)$, which has the pre-image $\mathcal{C}^{-1}(x, a, y) = \{(x, a, y_0, y_1) : ay_0 + (1 - a)y_1 = y\}$. Then the distribution induced on $(X, A, Y)$ is given by $\mathbb{P} = \mathbb{P}^* \circ \mathcal{C}^{-1}$ (as measures). That is, the data distribution $\mathbb{P}$ is given by taking a draw from $\mathbb{P}^*$ and coarsening it by $\mathcal{C}$. The data then consists of $n$ independent draws, $(X_i, A_i, Y_i) \sim \mathbb{P}$ for $1 \leq i \leq n$. We use $\mathbb{E}^*, \mathbb{E}$ to denote expectations with respect to $\mathbb{P}^*, \mathbb{P}$, respectively. Note that both have the same $(X, A)$-distribution, so anything involving only $(X, A)$ will be the same. For any function $f$ of $X, A, Y$ we let $\|f\|_p$ be the $L_p$-norm with respect to $\mathbb{P}$.

We assume throughout that all endogeneity is explained by $X$, known as unconfoundedness: $\mathbb{P}^*(A = 1 \mid X) = \mathbb{P}^*(A = 1 \mid Y^*(0), X) = \mathbb{P}^*(A = 1 \mid Y^*(1), X)$. That is, controlling for $X$, treatment assignment is independent of an individual's idiosyncrasies as relevant to their potential outcomes. Randomized experiments ensure this by design, often with $X \perp\!\!\!\perp A$ (complete randomization). Our results extend to observational settings assuming unconfoundedness, which is a common assumption in this setting [27]. For our purposes, the only technical difference between the experimental and observational settings is whether or not we exactly know the **propensity score**, defined as $e(X) = \mathbb{P}(A = 1 \mid X)$. We assume throughout that $e(X) \in (0, 1)$ almost surely, known as overlap. Note that assuming $Y = Y^*(A)$ also encapsulates non-interference [50].

In the following it will also be useful to define the **conditional mean of potential outcomes**:

$$\mu(X, a) = \mathbb{E}^*[Y^*(a) \mid X] = \mathbb{E}[Y \mid X, A = a], \quad a = 0, 1, \tag{1}$$

where the last equality follows from unconfoundedness and overlap, showing $\mu$ only depends on $\mathbb{P}$. We also define the **conditional average treatment effect (CATE) and sum (CATS) functions** as

$$\tau_-(X) = \mathbb{E}^*[\text{ITE} \mid X] = \mu(X, 1) - \mu(X, 0), \quad \tau_+(X) = \mu(X, 1) + \mu(X, 0). \tag{2}$$

## 2.1 Parameters of Interest

The primary parameter of interest we consider is the **fraction negatively affected (FNA)** by a wholesale change from treatment 0 to treatment 1:

$$\text{FNA} = \mathbb{P}^*(\text{ITE} < 0) = \mathbb{P}^*(Y^*(0) = 1, Y^*(1) = 0).$$

In particular, this quantity stands in contrast to the average treatment effect, $\text{ATE} = \mathbb{E}^*[\text{ITE}] = \mathbb{E}^*[Y^*(1)] - \mathbb{E}^*[Y^*(0)]$. A zero or positive ATE is generally interpreted to indicate a neutral or favorable treatment. However, having $\text{ATE} \geq 0$ need not mean having $\text{FNA} = 0$. A simple example is $(Y(0), Y(1)) \sim \text{Unif}(\{0, 1\}^2)$, which has $\text{ATE} = 0$ but $\text{FNA} = 1/4$.

Given we observe baseline covariates, we can more generally consider personalized treatment policies that assign treatments based on the value of $X$. Let $\pi_0, \pi_1 : \mathcal{X} \to \{0, 1\}$ denote any two such policies and let us define the fraction harmed by a change from $\pi_0$ to $\pi_1$ as

$$\text{FNA}_{\pi_0 \to \pi_1} = \mathbb{P}^*((\pi_1(X) - \pi_0(X)) \cdot \text{ITE} < 0) = \mathbb{P}^*(Y^*(\pi_0(X)) = 1, Y^*(\pi_1(X)) = 0).$$

This more general construct gives rise to several parameters of interest:

1. First, we recover our primary parameter of interest, $\text{FNA} = \text{FNA}_{0 \to 1}$, where 0 (or, 1) stands for the constant policy taking value 0 (or, 1) everywhere.

2. Second, given a proposed personalized treatment policy $\pi : \mathcal{X} \to \{0, 1\}$, the quantity $\text{FNA}_{0 \to \pi}$ is the fraction negatively affected by changing from a status quo of 0 to deploying the proposed policy $\pi$, which intervenes to treat the subpopulation with covariates $X$ such that $\pi(X) = 1$.

3. Third, we can recover the misclassification rate of the policy $\pi$:

$$\text{FNA}_{1-\pi \to \pi} = \mathbb{P}^*(\pi(X) \notin \text{argmax}_{a \in \{0,1\}} Y^*(a)).$$

This represents the fraction of individuals negatively affected by being misclassified by $\pi$: they should be treated by one treatment but are being given the other.

4. Of particular interest is the misclassification rate of the *optimal* personalized treatment policy, which assigns the treatment with larger conditional average outcome:

$$\pi^*(X) \in \text{argmax}_{a \in \{0,1\}} \mathbb{E}^*[Y^*(a) \mid X].$$

We are interested in $\text{FNA}_{1-\pi^* \to \pi^*}$. The policy $\pi^*$ can equivalently be characterized as the policy *maximizing* social welfare, $\mathbb{E}^*[Y^*(\pi(X))]$, or as *minimizing* misclassification, $\text{FNA}_{1-\pi \to \pi}$, both over *all* policies $\pi$. Given this, one hopes but cannot rule out that $\pi^*$ avoids negative impact.

## 3 Sharp Bounds

In this section, we derive sharp bounds on our parameters of interest. For the sake of generality we focus on the parameter $\text{FNA}_{\pi_0 \to \pi_1}$ as it captures all of our parameters of interest using different

$\pi_0, \pi_1$. The sharp bounds describe the set of values that $\text{FNA}_{\pi_0 \to \pi_1}$ can take if all we know is the distribution of the data, $\mathbb{P}$. This characterizes the most we can hope to learn at the limit of infinite data since this distribution is the most we can learn from draws from it, that is, our data.

To define this formally, recall that $\mathbb{P} = \mathbb{P}^* \circ \mathcal{C}^{-1}$ is given by $\mathbb{P}^*$ via coarsening. However, this mapping from $\mathbb{P}^*$ to $\mathbb{P}$ need not be invertible. Given a parameter $\psi(\mathbb{P}^*)$ that depends on the full distribution $\mathbb{P}^*$, we define its *identified set* as the set of all values consistent with $\mathbb{P}$, that is, that could be explained by some $\mathbb{P}^*$ that gives rise to the given data distribution and satisfies unconfoundedness:

$$\mathcal{S}(\psi; \mathbb{P}) = \big\{ \psi(\mathbb{P}^*) \ : \ \mathbb{P}^* \circ \mathcal{C}^{-1} = \mathbb{P}, \ \mathbb{P}^*(A = 1 \mid X) = \mathbb{P}^*(A = 1 \mid X, Y^*(a)), \ a = 0, 1 \big\}.$$

We are interested in the identified set of and sharp bounds on the parameter $\text{FNA}_{\pi_0 \to \pi_1}$.

When $|\mathcal{S}(\psi; \mathbb{P})| = 1$, we say that $\psi$ is *identifiable* because it is uniquely specified by the distribution of the data. Otherwise, we say it is *unidentifiable*: even given infinite data (equivalently, the distribution of the data), we cannot determine the value of $\psi(\mathbb{P}^*)$ based on the data alone since many values are consistent with the observations. For example, ATE *is* identifiable because $\text{ATE} = \mathbb{E}[\mu(X, 1)] - \mathbb{E}[\mu(X, 0)]$ and Eq. (1) shows that $\mu(X, a) = \mathbb{E}[Y \mid X, A = a]$ depends on $\mathbb{P}$ alone.

Is $\text{FNA}_{\pi_0 \to \pi_1}$ identifiable? Our first result characterizes exactly when, showing that generally, *no*.

**Theorem 1** (Identifiability). $\text{FNA}_{\pi_0 \to \pi_1}$ *is identifiable if and only if, for almost all $X$, either* $\pi_0(X) = \pi_1(X)$ *or* $\text{Var}(Y \mid X, A = 0) = 0$ *or* $\text{Var}(Y \mid X, A = 1) = 0$.

Thm. 1 shows that only in degenerate cases is $\text{FNA}_{\pi_0 \to \pi_1}$ identifiable. Generally, there will exist a non-negligible (positive probability) set of $X$'s where the policies $\pi_0, \pi_1$ differ and neither conditional outcome distribution is constant. Then, by Thm. 1, $\text{FNA}_{\pi_0 \to \pi_1}$ would be unidentifiable.

We are now prepared to state our main theorem characterizing the sharp bounds on $\text{FNA}_{\pi_0 \to \pi_1}$. Since $\text{FNA}_{\pi_0 \to \pi_1}$ is generally unidentifiable, the best we can do using the data is measure the set $\mathcal{S}(\text{FNA}_{\pi_0 \to \pi_1}; \mathbb{P})$. The *sharp lower and upper bounds* are $\inf(\mathcal{S}(\text{FNA}_{\pi_0 \to \pi_1}; \mathbb{P}))$ and $\sup(\mathcal{S}(\text{FNA}_{\pi_0 \to \pi_1}; \mathbb{P}))$, respectively, as these are precisely the largest (resp., smallest) lower (resp., upper) bounds on the set of possible values. The following result gives the sharp bounds and moreover shows the identified set is closed and convex so it is in fact an interval with the bounds as its endpoints.

**Theorem 2** (Sharp Bounds on FNA). *Fix any $\pi_0, \pi_1 : \mathcal{X} \to \{0, 1\}$. Set*

$$\text{FNA}_{\pi_0 \to \pi_1}^- = \mathbb{E}[\max\{(\pi_0(X) - \pi_1(X))\tau_-(X), 0\}], \tag{3}$$

$$\text{FNA}_{\pi_0 \to \pi_1}^+ = \mathbb{E}[\min\{\pi_1(X)(1 - \pi_0(X))\mu(X, 0) + \pi_0(X)(1 - \pi_1(X))(1 - \mu(X, 0)), \tag{4}$$
$$\pi_1(X)(1 - \pi_0(X))(1 - \mu(X, 1)) + \pi_0(X)(1 - \pi_1(X))\mu(X, 1)\}].$$

*Then, the identified set is the interval $\mathcal{S}(\text{FNA}_{\pi_0 \to \pi_1}; \mathbb{P}) = [\text{FNA}_{\pi_0 \to \pi_1}^-, \text{FNA}_{\pi_0 \to \pi_1}^+]$.*

The proof of Thm. 2 proceeds by applying the Fréchet-Hoeffding bounds [21, 51] for each level of $X$ and showing this remains sharp. Eqs. (3) and (4) simplify a lot for the change from all-0 to all-1:

$$\text{FNA}^- = -\mathbb{E}[\min\{\tau_-(X), 0\}], \quad \text{FNA}^+ = \mathbb{E}[\min\{\mu(X, 0), 1 - \mu(X, 1)\}]. \tag{5}$$

We can also simplify when considering the optimal policy. Plugging in the optimal policy we get:

$$\text{FNA}_{1 - \pi^* \to \pi^*}^- = 0, \ \text{FNA}_{1 - \pi^* \to \pi^*}^+ = \mathbb{E}[\min\{\mu(X, 0), 1 - \mu(X, 0), \mu(X, 1), 1 - \mu(X, 1)\}]. \tag{6}$$

Unlike most policies, it is always plausible that the optimal policy affects no one negatively: because $\text{FNA}_{1 - \pi^* \to \pi^*}^- = 0$, we always have $0 \in \mathcal{S}(\text{FNA}_{1 - \pi^* \to \pi^*}; \mathbb{P})$. On the other hand, it is generally also plausible that it does, that is, $\text{FNA}_{1 - \pi^* \to \pi^*}^+$ is generally nonzero. In agreement with Thm. 1, it is in fact zero *only* when $\mu(X, 0) \in \{0, 1\}$ or $\mu(X, 1) \in \{0, 1\}$ for almost all $X$.

**Remark 1** (The Non-Binary Case). We can generalize Thm. 2 to the non-binary-outcome case as well. Consider now *general* scalar outcomes $Y^*(0), Y^*(1) \in \mathbb{R}$, whether continuous, discrete, or mixed. Define the general parameter:

$$\psi_{\zeta, \delta}(\mathbb{P}^*) = \mathbb{P}^*(\zeta(X)\text{ITE} < \delta).$$

Then, $\text{FNA} = \mathbb{P}^*(Y^*(1) < Y^*(0)) = \psi_{1,0}$, $\text{FNA}_{\pi_0 \to \pi_1} = \psi_{\pi_1 - \pi_0, 0}$, and $\text{FNA}_{1 - \pi \to \pi} = \psi_{2\pi - 1, 0}$.

**Theorem 3.** *Fix $\zeta, \delta$. We then have*

$$\sup(\mathcal{S}(\psi_{\zeta, \delta}; \mathbb{P})) = 1 + \mathbb{E}\inf_y(\mathbb{P}(\zeta(X)Y < y + \delta \mid X, A = 1) - \mathbb{P}(\zeta(X)Y \le y \mid X, A = 0)),$$

$$\inf(\mathcal{S}(\psi_{\zeta, \delta}; \mathbb{P})) = \mathbb{E}\sup_y(\mathbb{P}(\zeta(X)Y < y + \delta \mid X, A = 1) - \mathbb{P}(\zeta(X)Y \le y \mid X, A = 0)).$$

We recover Eqs. (3) and (4) by min/maximizing over $y \in \{-1, 0, 0.5, 1\}$. For inference, we focus on the binary case. The non-binary case can be handled similarly but is more complicated as it requires learning the functions $y(X)$ that optimize the above $\inf$ and $\sup$ for each $X$. (Nonetheless, even if we learn the wrong functions we will get valid, albeit not sharp, bounds; see Thm. 7 and Lemma 6.)

**Remark 2** (Tail Expectations of Individual Treatment Effects). Kallus [28] considers bounds on the conditional value at risk (CVaR) of individual treatment effects (ITEs): for $\alpha \in (0, 1]$,

$$\mathrm{CVaR}_\alpha(\mathrm{ITE}) = \sup_\beta \mathbb{E}^* \big[ \beta + \alpha^{-1} \min\{\mathrm{ITE} - \beta, 0\} \big].$$

This is equal to the smallest subgroup-ATE among all $\alpha$-sized fractions of the population, that is, the average effect on the $(100\alpha)\%$-worst affected. It is generally unidentifiable from data. While Kallus [28] considers general real-valued outcomes, in the special case of binary outcomes considered here, $\mathrm{CVaR}_\alpha(\mathrm{ITE})$ is a function of just FNA and ATE:

**Lemma 1.** For $\alpha \in (0, 1)$, $\mathrm{CVaR}_\alpha(\mathrm{ITE}) = \max\{-1, -\alpha^{-1}\mathrm{FNA}, 1 - \alpha^{-1}(\mathrm{ATE} - 1)\}$.

As a consequence of Thm. 2 and Lemma 1, we immediately have that the identified set for $\mathrm{CVaR}_\alpha(\mathrm{ITE})$ for $\alpha \in (0, 1)$ is the interval $\mathcal{S}(\mathrm{CVaR}_\alpha(\mathrm{ITE}); \mathbb{P}) = [\max\{-1, -\alpha^{-1}\mathrm{FNA}^+, 1 - \alpha^{-1}(\mathrm{ATE} - 1)\}, \max\{-1, -\alpha^{-1}\mathrm{FNA}^-, 1 - \alpha^{-1}(\mathrm{ATE} - 1)\}]$, with $\mathrm{FNA}^\pm$ as defined in Eq. (5). Consequently, the endpoints are the sharp lower and upper bounds on $\mathrm{CVaR}_\alpha(\mathrm{ITE})$.

Kallus [28] gives the upper bound $\mathrm{CVaR}_\alpha(\mathrm{ITE}) \le \mathrm{CVaR}_\alpha(\tau_-(X))$ and shows it is tight given only $\tau_-(X)$ in the sense that is always realizable by some $\mathbb{P}^*$ with the same distribution of $\tau_-(X)$. However, this bound is generally not sharp, that is, it need not be realizable given $\mathbb{P}$, which characterizes more than just $\tau_-(X)$. In contrast, plugging $\mathrm{FNA}^-$ into Lemma 1 gives the sharp upper bound in the special case of binary outcomes, that is, this bound fully uses *all* the information in $\mathbb{P}$. Moreover, we have a finite sharp lower bound, whereas Kallus [28] shows that no lower bound exists when given only $\tau_-(X)$. Nonetheless, Kallus [28] crucially handles general real-valued outcomes. In particular, the CVaR curve as $\alpha$ varies is most interesting in settings with non-binary outcomes, since in binary settings, two numbers – the ATE and FNA – summarize all relevant information per Lemma 1.

## 4 Inference Methodology

In the previous section we characterized the sharp (meaning, tightest possible) bounds on FNA given the distribution of the data, that is, the best we could do if we had infinite data. We now address how to estimate these from actual data and characterize the uncertainty. That is, how to do inference on the parameters $\mathrm{FNA}^\pm_{\pi_0 \to \pi_1}$. Recall our data is $n$ independent draws $(X_i, A_i, Y_i) \sim \mathbb{P}$ for $i = 1, \ldots, n$.

In order to systematically handle all of parameters of interest, we will develop a generic, robust method for a large class of parameters that include all of our parameters of interest as special cases.

### 4.1 Average Hinge Effects

We consider a class parameters we call Average Hinge Effects (AHEs), parametrized by a positive integer $m \in \mathbb{N}$, a vector of signs $\rho \in \{-1, +1\}^m$, and a set of $3(m + 1)$ functions $g_\ell(x) = (g_\ell^{(0)}(x), g_\ell^{(1)}(x), g_\ell^{(2)}(x)) \in [-1, 1]^3$ for $\ell = 0, \ldots, m$:

$$\mathrm{AHE}^\rho_{g_0, \ldots, g_m} = \mathbb{E}\big[ g_0^{(0)}(X)\mu(X, 0) + g_0^{(1)}(X)\mu(X, 1) + g_0^{(2)}(X) \tag{7}$$
$$+ \sum_{\ell=1}^m \rho_\ell \min\{0, \, g_\ell^{(0)}(X)\mu(X, 0) + g_\ell^{(1)}(X)\mu(X, 1) + g_\ell^{(2)}(X)\}\big].$$

The AHE is so called because $\min\{0, x\}$ is called the hinge function. AHEs cover all of our parameters of interest: $\mathrm{FNA}^- = \mathrm{AHE}^{(-)}_{(0,0,0),(-1,1,0)}$, $\mathrm{FNA}^+ = \mathrm{AHE}^{(+)}_{(1,0,0),(-1,-1,1)}$, $\mathrm{FNA}^-_{\pi_0 \to \pi_1} = \mathrm{AHE}^{(-)}_{(0,0,0),(\pi_0 - \pi_1, \pi_1 - \pi_0, 0)}$, $\mathrm{FNA}^+_{\pi_0 \to \pi_1} = \mathrm{AHE}^{(+)}_{(\pi_0(1-\pi_1), \pi_1 - \pi_0, 0), (\pi_0 - \pi_1, \pi_0 - \pi_1, -1)}$, and $\mathrm{FNA}^+_{1-\pi^* \to \pi^*} = \mathrm{AHE}^{(+,+)}_{(1,0,0),(-1,1,0),(-1,-1,1)}$. For now, we focus on any given $m, \rho, g_0, \ldots, g_m$.

### 4.2 Re-formulating the AHE

The AHE is defined as an average of a function of $X$. The only unknown part of this function is $\mu(x, a)$. A naïve approach would be to estimate $\mu$ by $\hat{\mu}$, plug it into Eq. (7) and replace $\mathbb{E}$ by a

sample average over $X_1, \ldots, X_n$. This estimate, however, will *not* behave like a sample average approximation of Eq. (7). In particular, it has a nonzero derivative in $\hat{\mu}$, so that the errors in $\hat{\mu}$ directly translate to errors in AHE estimation. For example, even if $\hat{\mu}$ converges as fast as $O_p(1/\sqrt{n})$, this will introduce significant errors on the same order as the convergence of sample averages, which can imperil both efficiency and inference. And, if it converges more slowly, as would generally be the case when using flexible machine learning methods for estimating (as parametric ones will inevitably be misspecified), it can even affect the convergence rate of the final estimate. Worst yet, if $\hat{\mu}$ is inconsistent for $\mu$ we have no hope of consistency for our estimate or an understanding of the direction of its bias. Generally, we would like to avoid depending on how exactly we fit $\hat{\mu}$, provide guarantees even when it is estimated slowly and regardless of what method is used to estimate it, and even provide guarantees when its estimation is inconsistent.

Our approach is based on focusing on an alternative formulation of AHE as a sample average where plugging in wrong values for some unknown functions does not affect the formulation too much.

Toward that end, let us first define $\eta_\ell : \mathcal{X} \to [-3, 3]$ by

$$\eta_\ell(x) = g_\ell^{(0)}(X)\mu(X, 0) + g_\ell^{(1)}(X)\mu(X, 1) + g_\ell^{(2)}(X). \tag{8}$$

Specifically, we have $\eta_1(X) = (\pi_1(X) - \pi_0(X))\tau_-(X)$ for $\mathrm{FNA}^-_{\pi_0 \to \pi_1}$, $\eta_1(X) = (\pi_1(X) - \pi_0(X))(1 - \tau_+(X))$ for $\mathrm{FNA}^+_{\pi_0 \to \pi_1}$, and $\eta_1(X) = \tau_-(X)$, $\eta_2(X) = 1 - \tau_+(X)$ for $\mathrm{FNA}^+_{1-\pi^* \to \pi^*}$.

Given $\breve{e}, \breve{\mu}, \breve{\eta}_1, \ldots, \breve{\eta}_m$, understood as (possibly wrong) stand-ins for $e, \mu, \eta_1, \ldots, \eta_m$, define

$$
\begin{aligned}
\phi^\rho_{g_0,\ldots,g_m}(X, A, Y; \breve{e}, \breve{\mu}, \breve{\eta}_1, \ldots, \breve{\eta}_m) = {} & g_0^{(2)}(X) + \textstyle\sum_{\ell=1}^m \rho_\ell \mathbb{I}[\breve{\eta}_\ell(X) \leq 0] g_\ell^{(2)}(X) \\
& + \left(g_0^{(0)}(X) + \textstyle\sum_{\ell=1}^m \rho_\ell \mathbb{I}[\breve{\eta}_\ell(X) \leq 0] g_\ell^{(0)}(X)\right) \frac{(A - \breve{e}(X))\breve{\mu}(X,0) + (1-A)Y}{1 - \breve{e}(X)} \\
& + \left(g_0^{(1)}(X) + \textstyle\sum_{\ell=1}^m \rho_\ell \mathbb{I}[\breve{\eta}_\ell(X) \leq 0] g_\ell^{(1)}(X)\right) \frac{(\breve{e}(X) - A)\breve{\mu}(X,1) + AY}{\breve{e}(X)}.
\end{aligned}
$$

If we plug in the correct values of the nuisance parameters $e, \mu, \eta_1, \ldots, \eta_m$, then we would have

$$\mathrm{AHE}^\rho_{g_0,\ldots,g_m} = \mathbb{E}[\phi^\rho_{g_0,\ldots,g_m}(X, A, Y; e, \mu, \eta_1, \ldots, \eta_m)]. \tag{9}$$

Crucially, unlike the definition of AHE as an average in Eq. (7), we will show in Sec. 5 that this new representation as an average remains faithful *even* when we make small errors in the nuisances.

### 4.3 Inference Algorithm

Given our special construction of $\phi^\rho_{g_0,\ldots,g_m}$, our algorithm, presented in Alg. 1, follows a simple recipe: we estimate the nuisances in a cross-fold fashion and plug them into Eq. (9), approximating the expectation by a sample average. In Sec. 5 we will show that the special structure of $\phi^\rho_{g_0,\ldots,g_m}$ affords this procedure a lot of robustness. The use of cross-fitting ensures nuisance estimates are independent of samples applied thereto [12, 52, 63]. This allows our analysis to only require lax rate assumptions on the nuisance fitting without requiring any additional regularity on restricting *how* the fitting is done. (If we assume nuisance estimates belong to a Donsker class with probability tending to 1, all of our results will hold even without cross-fitting; we omit this option for brevity.)

### 4.4 Nuisance Fitting

Our algorithm requires methods for fitting the nuisances $e, \mu, \eta_1, \ldots, \eta_m$. Fitting $e$ and $\mu$ amounts to binary regressions (probabilistic classification), which can be done with any of a variety of standard supervised learning methods, whether logistic regression, random forests, or neural nets.

There are different ways to fit $\eta_\ell$. In particular, despite the fact that $\eta_\ell$ is determined by $\mu$, we treat it as a separate nuisance to allow for specialized learners. Of course, the simplest way to fit it is to simply plugin an estimate for $\mu$ into the definition of $\eta_\ell$ in Eq. (8), and that is a legitimate option. For all of our parameters of interest, all $\eta_\ell$ nuisances are given by learning just two functions: $\tau_-$ and $\tau_+$ as in Eq. (2). Therefore, we may alternatively learn these directly and plug them into $\eta_\ell$. For example, there exists a wide literature specialized to learning $\tau_-$ motivated by the observation that effect signals are often more nuanced and can therefore be washed out by the noise in baselines if we simply difference $\mu$-estimates [3, 26, 34, 37, 46, 59]. We can similarly apply of all these approaches to learning $\tau_+$ by simply first mutating the outcome data as $Y \leftarrow (2A - 1)Y$. Notably, [34, 46] give rates of convergence for $\tau_-$-learning, which we can use to satisfy our assumptions in the next section.

**Algorithm 1** Point estimate and CIs for $\mathrm{AHE}^\rho_{g_0,\ldots,g_m}$

---

**Input:** Data $\{(X_i, A_i, Y_i) : i = 1, \ldots, n\}$, number of folds $K$, estimators for $e, \mu, \eta_1, \ldots, \eta_m$
1: **for** $k = 1, \ldots, K$ **do**
2:     Estimate $\hat{e}^{(k)}, \hat{\mu}^{(k)}, \hat{\eta}_1^{(k)}, \ldots, \hat{\eta}_m^{(k)}$ using data $\{(X_i, A_i, Y_i) : i \not\equiv k - 1 \pmod{K}\}$
3:     **for** $i \equiv k - 1 \pmod{K}$ **do** set $\phi_i = \phi^\rho_{g_0,\ldots,g_m}(X_i, A_i, Y_i; \hat{e}^{(k)}, \hat{\mu}^{(k)}, \hat{\eta}_1^{(k)}, \ldots, \hat{\eta}_m^{(k)})$
4: **end for**
5: Set $\widehat{\mathrm{AHE}}^\rho_{g_0,\ldots,g_m} = \frac{1}{n} \sum_{i=1}^n \phi_i$, $\hat{\mathrm{se}} = \sqrt{\frac{1}{n(n-1)} \sum_{i=1}^n (\phi_i - \widehat{\mathrm{AHE}}^\rho_{g_0,\ldots,g_m})^2}$
6: Return point estimate $\widehat{\mathrm{AHE}}^\rho_{g_0,\ldots,g_m}$ and $\beta$-CIs $[\widehat{\mathrm{AHE}}^\rho_{g_0,\ldots,g_m} \pm \Phi^{-1}((1+\beta)/2)\hat{\mathrm{se}}]$

---

# 5 Robustness Guarantees for Inference

We now show our inference method has some nice robustness guarantees. This will depend on showing that using our special $\phi$ renders Eq. (9) insensitive to errors in the nuisances. The specific level of errors allowed will depend on the *sharpness* of margin satisfied by $\eta_\ell$, as defined below.

**Definition 1.** We say that a given function $f : \mathcal{X} \to \mathbb{R}$ satisfies a margin with sharpness $\alpha \in [0, \infty]$ (or, an $\alpha$-margin) if there exists $t_0 > 0$ such that $\mathbb{P}(0 < |f(X)| \leq t) \leq (t/t_0)^\alpha$ for all $t > 0$. (We use the convention that $x^\infty = 0$ if $0 \leq x < 1$, $1^\infty = 1$, and $x^\infty = \infty$ if $x > 1$.)

Every $f$ trivially satisfies a 0-margin, but usually a sharper margin holds. If $f(X)$ is almost surely either zero or bounded away from zero, then $f$ satisfies an infinitely sharp margin:

**Lemma 2.** *If* $\mathbb{P}(|f(X)| \geq t_0) = \mathbb{P}(f(X) \neq 0)$ *for some* $t_0 > 0$ *then* $f$ *satisfies an* $\infty$-*margin.*

In particular, if $X$ has finite support, then we can always set $t_0 = \inf\{|f(x)| : x \in \mathcal{X}, f(x) \neq 0\}$. But, this also works more generally. Consider $\eta_1(X) = (\pi_1(X) - \pi_0(X))\tau_-(X)$ for $\mathrm{FNA}^-_{\pi_0 \to \pi_1}$. If $\inf\{|\tau_-(x)| : \pi_1(x) \neq \pi_0(x), \tau_-(x) \neq 0\} > 0$, that is, the policies only differ either far from the boundary where $\tau_-$ is 0 or exactly on this boundary, then Lemma 2 ensures sharpness $\infty$.

More generally, we should expect a margin with sharpness 1 in continuous settings.

**Lemma 3.** *If the cumulative distribution function (CDF) of* $f(X)$ *is boundedly differentiable on* $(-\epsilon, 0) \cup (0, \epsilon)$ *for some* $\epsilon > 0$, *then* $f$ *satisfies a 1-margin.*

Consider again the example of $\eta_1(X) = (\pi_1(X) - \pi_0(X))\tau_-(X)$. If $\tau_-(X)$ is a continuous random variable with a bounded density near 0, then Lemma 3 ensures a margin with sharpness 1. This holds for *any* two policies, including $\pi_1 = 1$, $\pi_0 = 0$, which differ everywhere. The same conclusion would also hold if the CDF of $\tau_-(X)$ was just continuously differentiable at 0.

Armed with the notion of margin, we can state in what sense the representation Eq. (9) is robust.

**Theorem 4.** *Fix any* $\tilde{e}, \check{e} : \mathcal{X} \to [\bar{e}, 1 - \bar{e}]$ *with* $\bar{e} > 0$, $\tilde{\mu}, \check{\mu} : \mathcal{X} \times \{0, 1\} \to [0, 1]$, $\tilde{\eta}_\ell, \check{\eta}_\ell : \mathcal{X} \to [-3, 3]$ *for* $\ell = 1, \ldots, m$. *Fix any* $p_1, \ldots, p_\ell \in [1, \infty]$. *Suppose that either* $\tilde{e} = e$ *or* $\tilde{\mu} = \mu$ *(or both). Set* $\kappa_\ell = 1$ *if* $\tilde{\eta}_\ell = \eta_\ell$ *and otherwise set* $\kappa_\ell = 0$. *Further assume that* $\mathbb{P}(\tilde{\eta}_\ell(X) = 0, \check{\eta}_\ell(X) \neq 0) = 0$. *Finally, suppose that* $\tilde{\eta}_\ell$ *satisfies a margin with sharpness* $\alpha_\ell$. *Then, for some constant* $c > 0$, *we have*

$$c\big|\mathbb{E}[\phi^\rho_{g_0,\ldots,g_m}(X, A, Y; \check{e}, \check{\mu}, \check{\eta}_1, \ldots, \check{\eta}_m)] - \mathbb{E}[\phi^\rho_{g_0,\ldots,g_m}(X, A, Y; \tilde{e}, \tilde{\mu}, \tilde{\eta}_1, \ldots, \tilde{\eta}_m)]\big|$$
$$\leq \|\check{e} - e\|_2 \|\check{\mu} - \mu\|_2 + \sum_{\ell=1}^m \|\check{\eta}_\ell - \tilde{\eta}_\ell\|_{p_\ell}^{\frac{p_\ell(\kappa_\ell + \alpha_\ell)}{p_\ell + \alpha_\ell}},$$

$$c\big\|\phi^\rho_{g_0,\ldots,g_m}(X, A, Y; \check{e}, \check{\mu}, \check{\eta}_1, \ldots, \check{\eta}_m) - \phi^\rho_{g_0,\ldots,g_m}(X, A, Y; \tilde{e}, \tilde{\mu}, \tilde{\eta}_1, \ldots, \tilde{\eta}_m)\big\|_2$$
$$\leq \|\check{e} - \tilde{e}\|_2 + \|\check{\mu} - \tilde{\mu}\|_2 + \sum_{\ell=1}^m \|\check{\eta}_\ell - \tilde{\eta}_\ell\|_{p_\ell}^{\frac{p\alpha_\ell}{2p + 2\alpha_\ell}}.$$

*In the above, we use the convention that* $\frac{\infty a + b}{\infty c + d} = \frac{a}{c}$.

## 5.1 Local Robustness, Confidence Intervals, and Efficiency

Thm. 4 has several important implications. Our first result shows that, if we estimate all nuisances correctly but potentially slowly at nonparametric rates, then we can ensure our estimator looks like a sample average approximation of Eq. (9) and our CIs are calibrated.

**Theorem 5.** *Suppose* $e(X) \in [\bar{e}, 1 - \bar{e}]$ *with* $\bar{e} > 0$, $\eta_\ell$ *satisfies an* $\alpha_\ell$-*margin, and for* $k = 1, \ldots, K$
$$\|\hat{e}^{(k)} - e\|_2 \|\hat{\mu}^{(k)} - \mu\|_2 = o_p(n^{-\frac{1}{2}}), \quad \|\hat{\eta}_\ell^{(k)} - \eta_\ell\|_{p_\ell} = o_p(n^{-\frac{p_\ell + \alpha_\ell}{2p_\ell(1 + \alpha_\ell)}}) \text{ with } p_\ell \in [1, \infty], \text{ and}$$

$\mathbb{P}(\bar{e} \le \hat{e}^{(k)}(x) \le 1 - \bar{e},\, 0 \le \hat{\mu}^{(k)}(x) \le 1,\, |\hat{\eta}_\ell^{(k)}(x)| \le 3\mathbb{I}[\eta_\ell(x) \ne 0],\, \ell \le m,\, a.e.\ x) \to 1.$ *Then,*

$$\widehat{\mathrm{AHE}}_{g_0,\ldots,g_m}^{\rho} = \tfrac{1}{n}\sum_{i=1}^{n} \phi_{g_0,\ldots,g_m}^{\rho}(X_i, A_i, Y_i; e, \mu, \eta_1, \ldots, \eta_m) + o_p(n^{-\frac{1}{2}}),$$

$$\mathbb{P}(\mathrm{AHE}_{g_0,\ldots,g_m}^{\rho} \in [\widehat{\mathrm{AHE}}_{g_0,\ldots,g_m}^{\rho} \pm \Phi^{-1}((1+\beta)/2)\dot{\mathrm{se}}]) \to \beta \quad \forall \beta \in (0,1).$$

The conditions of Thm. 5 allow that we learn $e, \mu, \eta_1, \ldots, \eta_m$ rather slowly. In particular, unlike naïvely plugging in a $\mu$-estimate into Eq. (7) and taking sample averages, we do not depend at all on *how* $\mu$ is estimated and the uncertainty therein, provided some slow nonparametric rates hold. For example, learning $e, \mu$ in with $o_p(n^{-1/4})$-rates in $L_2$-error suffices. Or, if $e$ is known, then consistently estimating $\mu$ *without* a rate suffices. For $\eta_\ell$, if an $\infty$-margin holds, it suffices to learn $\eta_\ell$ consistently (no rate) in $L_\infty$-error or at $o_p(n^{-1/4})$-rates in $L_2$-error. If a 1-margin holds, it suffices to learn at $o_p(n^{-1/4})$-rates in $L_\infty$-error. While conditions like metric entropy imply $L_2$-error rates [60], for many classes like smooth functions the $L_p$ error rates are even the same for all $p \in [1, \infty]$ [54].

In fact, under one more condition, our estimator achieves the semiparametric efficiency lower bound.

**Theorem 6.** *Suppose* $\mathbb{P}(\eta_\ell(X) = 0, g_\ell^{(a)}(X) \ne 0) = 0$ *for* $\ell = 1, \ldots, m$, $a = 0, 1$. *The semiparametric efficiency lower bound for* $\mathrm{AHE}_{g_0,\ldots,g_m}^{\rho}$ *is* $\mathrm{Var}(\phi_{g_0,\ldots,g_m}^{\rho}(X, A, Y; e, \mu, \eta_1, \ldots, \eta_m))$, *whether or not* $e(X)$ *is unknown (varies in the model) or known (fixed in the model).*

**Corollary 1.** *Under the assumptions of Thms. 5 and 6,* $\widehat{\mathrm{AHE}}_{g_0,\ldots,g_m}^{\rho}$ *is efficient: it is regular and achieves the minimum asymptotic variance among all regular estimators.*

For all of our parameters of interest, the condition of Thm. 6 holds if $\mathbb{P}(\tau_-(X) = 0) = \mathbb{P}(\tau_+(X) = 0) = 0$ (*e.g.*, $\tau_-(X), \tau_+(X)$ are continuous). Similarly, the condition that $|\hat{\eta}_\ell^{(k)}(x)| \le 3\mathbb{I}[\eta_\ell(x) \ne 0]$ in Thm. 5 is immediately satisfied if $\hat{\tau}_-^{(k)}(X), \hat{\tau}_+^{(k)}(X)$ are also almost never 0 (*e.g.*, continuous).

## 5.2 Double Robustness and Double Validity

Next we show that Alg. 1 is very robust to incorrectly learning some nuisances.

**Theorem 7.** *Fix any* $\tilde{\mu}, \tilde{e}, \tilde{\eta}_1, \ldots, \tilde{\eta}_m$ *with* $\tilde{\mu}(X) \in [0,1]$, $\tilde{e}(X) \in [\bar{e}, 1 - \bar{e}]$ *with* $\bar{e} > 0$. *Set* $\kappa_\ell = 1$ *if* $\tilde{\eta}_\ell = \eta_\ell$ *and otherwise set* $\kappa_\ell = 0$. *Suppose* $\tilde{\eta}_\ell$ *satisfies a margin with sharpness* $\alpha_\ell$, *and for* $k = 1, \ldots, K$ *that* $\|\hat{e}^{(k)} - \tilde{e}\|_2 = o_p(1)$, $\|\hat{\mu}^{(k)} - \tilde{\mu}\|_2 = o_p(1)$, $\|\hat{\eta}_\ell^{(k)} - \tilde{\eta}_\ell\|_{p_\ell} = O_p(n^{-\frac{p_\ell + \alpha_\ell}{2p_\ell(\kappa_\ell + \alpha_\ell)}})$ *for some* $p_\ell \in [1, \infty]$, *and* $\mathbb{P}(\bar{e} \le \hat{e}^{(k)}(x) \le 1 - \bar{e},\, 0 \le \hat{\mu}^{(k)}(x) \le 1,\, |\hat{\eta}_\ell^{(k)}(x)| \le 3\mathbb{I}[\tilde{\eta}_\ell(x) \ne 0],\, \ell \le m,\, a.e.\ x) \to 1$. *If either* $\|\hat{e}^{(k)} - e\|_2 = O_p(n^{-1/2})$ *or* $\|\hat{\mu}^{(k)} - \mu\|_2 = O_p(n^{-1/2})$, *then*

$$\widehat{\mathrm{AHE}}_{g_0,\ldots,g_m}^{\rho} = \mathrm{AHE}_{g_0,\ldots,g_m}^{\rho} + O_p(n^{-\frac{1}{2}}) \text{ if } \kappa_1 = \cdots = \kappa_m = 1, \qquad \textit{(double robustness)}$$

$$\widehat{\mathrm{AHE}}_{g_0,\ldots,g_m}^{\rho} \ge \mathrm{AHE}_{g_0,\ldots,g_m}^{\rho} + O_p(n^{-\frac{1}{2}}) \text{ if } \rho_\ell = + \text{ whenever } \kappa_\ell = 0, \quad \textit{(double validity, upper)}$$

$$\widehat{\mathrm{AHE}}_{g_0,\ldots,g_m}^{\rho} \le \mathrm{AHE}_{g_0,\ldots,g_m}^{\rho} + O_p(n^{-\frac{1}{2}}) \text{ if } \rho_\ell = - \text{ whenever } \kappa_\ell = 0. \quad \textit{(double validity, lower)}$$

The first equation (double robustness) in Thm. 7 shows our estimator remain consistent for the AHE even if we *incorrectly* learn either $e$ or $\mu$, bot not both, as long as we correctly learn $\eta_1, \ldots, \eta_m$.

The second two equations (double validity) in Thm. 7 show what happens when we also learn $\eta_\ell$ *incorrectly*. For upper double validity (second equation), we show that, as long as the $\eta_\ell$ that are misestimated correspond to convex terms in the AHE ($\rho_\ell = +$), we remain consistent for an *upper bound* on the AHE, even if we *also* incorrectly learn either $e$ or $\mu$, bot not both. In particular, for $\mathrm{FNA}^+$ we only have convex terms, so we are guaranteed and upper bound on the upper bound, that is, we still estimate a valid upper bound on FNA when we misestimate the CATE $\tau_-$ (which is $\eta_1$ for $\mathrm{FNA}^+$) and one of $e$ or $\mu$, it just may not be sharp. We also have a symmetric result for lower double validity (third equation), and applying it to $\mathrm{FNA}^-$ we find that we will still estimate a valid upper bound, albeit possibly unsharp, on FNA when we misestimate $\tau_+$ and one of $e$ or $\mu$. The significance is that we can really trust the results of Alg. 1 are truly bounds on FNA, even if we make mistakes, and therefore conclusions about harm based on our estimates and inferences can be highly credible.

# 6 Empirical Investigation

We demonstrate our method in a simulation and a case study with data from a real experiment. Replication code is available at `https://github.com/CausalML/BoundsOnFractionNegativelyAffected`. Experiments were run on AWS c5.24xlarge.

**Simulation Study** We consider 7-dimensional standard normal $X$ and set $\mu(x, a) = (1 + \exp(-\beta(2\mathbb{I}[\xi(x)x_2 > 0] - 1)(\mathbb{I}[\xi(x)x_1 \leq 0] + (2a - 1)\mathbb{I}[\xi(x)x_1 > 0])))^{-1}$, where $\beta \geq 0$ and $\xi(x) = 2\operatorname{XOR}(\mathbb{I}[x_3 > 0], \ldots, \mathbb{I}[x_7 > 0]) - 1$. Here $\beta$ controls how much $X$ predicts $Y(a)$. The dashed lines in Fig. 1 show the sharp bounds of Thm. 2 for FNA and $\mathrm{FNA}_{1-\pi^* \to \pi^*}$ as we vary $\beta$. We can see that, as $X$ becomes more predictive, the bounds go from $[0, 0.5]$ to a point at $0.25$. Note there is no *true* value for FNA as we are not specifying the joint distribution of potential outcomes; instead, per Thm. 2, there are joint distributions with FNA equal to any point between the bounds.

Next, we estimate these true bounds. We fix $e(X) = (1+\exp(0.25-\mathbb{I}[x_3 > 0]+0.5\mathbb{I}[x_4 > 0])^{-1}$, and draw $n$ samples from $X \sim \mathcal{N}(0, I)$, $A \mid X \sim \operatorname{Bernoulli}(e(X))$, $Y \mid X, A \sim \operatorname{Bernoulli}(\mu(X, A))$. We apply Alg. 1 to this data with $K = 5$, estimating $e, \mu$ using random regression forests and $\tau_-, \tau_+$ using causal forests (all using *R* package `grf` with default parameters). The solid lines in Fig. 1 show the point estimate and 95%-CI output by Alg. 1 for a single draw of $n = 12800$ data for each $\beta$. In Figs. 2 and 3 we plot the average root mean squared error of point estimates and coverage of CIs over 100 replications with $\beta = 3$ and varying $n$. We compare the performance to the (cross-fitted) "plugin" estimator that just plugs in the (same) estimates for $\mu$ into Eq. (7) and approximate the expectation by a sample average. Shaded intervals denote 95%-CIs for performance (*i.e.*, due to finite replications).

**Case Study** Using data from [6] (BSD license), we compare three different assistance programs offered to French unemployed individuals: the standard benefits (sta), access to public-run counseling (pub), and to private-run counseling (pri). Our binary outcome is reemployment within six months. For example, we can consider a hypothetical scenario where we change from a private to a public counseling provider. As reported by [6], the ATE for this hypothetically change is slightly positive, so a policymaker might therefore enact it, especially if, for example, it offered cost savings or other operational benefits. We consider who may be harmed by this: how many who remain unemployed under the new public program that would have been reemployed under the old private program.

For any pair of arms, we consider the sharp bounds on the FNA from one to other or the other way and on the misclassification rate of the optimal policy choosing between the two. Following Kallus [28], we set $X$ to all pre-treatment covariates in table 2 of Behaghel et al. [6]., and we consider the sharp bounds either with these $X$ or given no $X$ at all. We fit $\mu$ using linear regression and $\tau_-, \tau_+$ using doubly-robust pseudo-outcome linear regression. Since the propensity is known, in light of the guarantees of Thm. 7, we do not worry about misspecifying $\mu$, only using it for variance reduction by accounting for main effects, and we do not worry about misspecifying $\tau_-, \tau_+$, only hoping to tighten the bounds some from using no $X$. The results are shown in Fig. 4. We see that, while covariates are somewhat uninformative, we are still able to tighten bounds compared to no $X$. In particular, for the change from sta to pri or pub or the change from pri to pub, without $X$ the lower bounds on FNA are 0, while with $X$ the lower bounds as well as the lower confidence bounds on these bounds are strictly positive. This means that, with $X$ and the methods developed we can prove that there must be some harm by this change, something we could not do otherwise and such a finding can bolster further work to investigate and address this harm.

# 7 Connections, Limitations, Extensions, and Conclusions

**Connections to partial identification** Partial identification of unknowable parameters has a long tradition in econometrics [11, 35, 40, 41, 43, 45, 53, 55]. Some approaches focus on average treatment effects in the presence of confounding [8, 13, 15, 17, 31, 49, 56, 62], among which [8, 13, 17, 62] are notable for conducting semiparametric inference on bounds, but for confounding rather than for individual effects, as we do. Some others, like us, focus on understanding the joint distribution of potential outcomes [1, 16, 18, 25, 42]. Like us, these heavily leverage the Fréchet-Hoeffding bounds [9, 21, 48, 51, 61]. Unlike these, we use efficient and robust inference that can aggregate across covariates to tighten bounds by leveraging machine learning of conditional-mean outcome functions.

**Connections to fairness** Understanding the distribution of impact is a central problem in algorithmic fairness [5, 44]. Here, we specifically focused on the *differential* impact of interventions,

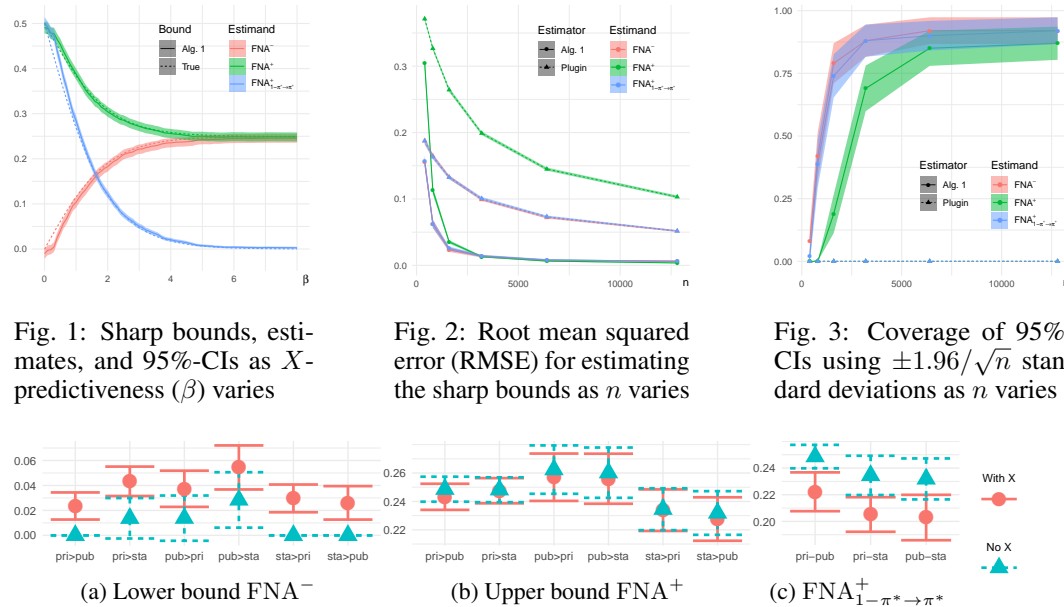

Fig. 1: Sharp bounds, estimates, and 95%-CIs as $X$-predictiveness ($\beta$) varies

Fig. 2: Root mean squared error (RMSE) for estimating the sharp bounds as $n$ varies

Fig. 3: Coverage of 95%-CIs using $\pm 1.96/\sqrt{n}$ standard deviations as $n$ varies

(a) Lower bound FNA$^-$

(b) Upper bound FNA$^+$

(c) FNA$^+_{1-\pi^* \to \pi^*}$

Fig. 4: Estimated sharp bounds for FNA in the case study with covariates and without covariates

which brought up issues of identification. In contrast, in algorithmic fairness one usually measures disparities in observed outcomes, such as in the form of the loss function of a model on a labeled example [2, 7, 22, 33]. A line of work specifically focuses on identification issues in algorithmic fairness [10, 14, 19, 29, 30, 32, 36, 38]. Notably, [32] also conduct semiparametric inference on sharp bounds derived from the Fréchet-Hoeffding bounds, but in the context of algorithm evaluation with unobserved protected labels rather than interventions with unobserved counterfactuals. And, [30] do consider unobserved counterfactuals in assessing equality of opportunity [24] for intervention-prioritization policies, but (appropriately in their own context) they assume away our primary focus here by assuming an a priori known bound on FNA (possibly zero, *i.e.*, monotone treatment response).

**Limitations and Extensions**     Restricting outcomes to be binary is one limitation of our work, and extensions to continuous settings is an interesting avenue of future research. Another limitation is that, while we can bound the FNA, it can still be hard to asses *who* is negatively affected. This is, unfortunately, impossible for the same reason FNA is unidentifiable, but a place to start such analyses may be to consider who are individuals with $\tau_-(X) \leq 0$ and even characterize that group via summary statistics compared to the population. Indeed, per Eq. (5), the lower bound on FNA is exactly the (negative of the) ATE on this group. Another important consideration is whether the data is representative: FNA refers to the fraction of the *studied population*, which might differ from the population of interest. *E.g.*, if an experiment did not enroll a representative sample, we may be systematically excluding some groups from consideration. If such unrepresentativeness is explained by covariates $X$ (*i.e.*, missing at random), the solution is simple: we reweight. If not explained by $X$ (*i.e.*, missing *not* at random), then we need to also account for this additional source of unidentifiability. An avenue for future research is to combine such ambiguity with counterfactual ambiguity. Another concern is whether the outcome represents the impact we want to measure [47].

**Conclusions**     Our tools support drawing credible conclusions about the potential negative impact of interventions: they both account for ambiguity due to unobserved counterfactuals (while mitigating it using covariates) as well as strongly guard against slow or inconsistent learning of necessary nuisances functions. Robust inference on the lower bounds, in particular, crucially provides watertight demonstrations of negative impact, which can bolster efforts to mitigate harm and improve equity.

# Acknowledgments

I am grateful for the helpful comments of the anonymous reviewers and for many insightful and thought-inspiring conversations with my colleagues at Netflix.

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
