**Supplementary Materials for**

# What's the Harm? Sharp Bounds on the Fraction Negatively Affected by Treatment
Nathan Kallus



## A  Proofs for Sec. 3

### A.1  Proof of Thm. 1

*Proof of Thm. 1.* It is easiest to prove this as a consequence of Thm. 2, since we already prove the latter below. By Thm. 2, the statement that $\mathrm{FNA}_{\pi_0 \to \pi_1}$ is identifiable is equivalent to the statement that $\mathrm{FNA}^+_{\pi_0 \to \pi_1} - \mathrm{FNA}^-_{\pi_0 \to \pi_1} = 0$, as defined in Eqs. (3) and (4). Note, moreover, that $\mathrm{FNA}^+_{\pi_0 \to \pi_1} - \mathrm{FNA}^-_{\pi_0 \to \pi_1} = \mathbb{E}[\nu(X)]$, where

$$
\begin{aligned}
\nu(X) &= \pi_1(X)(1 - \pi_0(X))(\min\{\mu(X, 0), 1 - \mu(X, 1)\} + \min\{\tau_-(X), 0\}) \\
&\quad + \pi_0(X)(1 - \pi_1(X))(\min\{\mu(X, 1), 1 - \mu(X, 0)\} + \min\{-\tau_-(X), 0\}) \\
&= (\pi_0(X) + \pi_1(X) - 2\pi_0(X)\pi_1(X))\min\{\mu(X, 1), 1 - \mu(X, 1), \mu(X, 0), 1 - \mu(X, 0)\}.
\end{aligned}
$$

and that $\nu(X) \geq 0$ is a nonnegative variable. Therefore, the statement that $\mathrm{FNA}_{\pi_0 \to \pi_1}$ is identifiable is equivalent to the statement that $\mathbb{P}(\nu(X) = 0) = 1$. From the above simplification of $\nu(X)$ and since $\mu(X, A) \in [0, 1]$, it is immediate that the event $\nu(X) = 0$ is equivalent to the $X$-measurable event $(\pi_1(X) = \pi_0(X)) \vee (\mu(X, 0) \in \{0, 1\}) \vee (\mu(X, 1) \in \{0, 1\})$. Noting that $\mathrm{Var}(Y \mid X, A = a) = 0$ is equivalent to $\mu(X, a) \in \{0, 1\}$ completes the proof. $\qquad\square$

### A.2  Proof of Thm. 2

*Proof.* By iterated expectations we can write

$$
\begin{aligned}
\mathrm{FNA}_{\pi_0 \to \pi_1} &= \mathbb{E}[\kappa(X)], \\
\text{where} \quad \kappa(X) &= \mathbb{P}^*(Y^*(\pi_0(X)) = 1, \, Y^*(\pi_1(X)) = 0 \mid X) \\
&= \pi_1(X)(1 - \pi_0(X))\mathbb{P}^*(Y(0) = 1, Y(1) = 0 \mid X) \\
&\quad + \pi_0(X)(1 - \pi_1(X))\mathbb{P}^*(Y(0) = 0, Y(1) = 1 \mid X).
\end{aligned}
$$

Let use first show that $\mathrm{FNA}^-_{\pi_0 \to \pi_1} \leq \inf(\mathcal{S}(\mathrm{FNA}_{\pi_0 \to \pi_1}; \mathbb{P}))$. Consider any feasible $\mathbb{P}^*$. By union bound, and since probabilities are in $[0, 1]$, we have

$$
\begin{aligned}
\mathbb{P}^*(Y(0) = 1, Y(1) = 0 \mid X) &= 1 - \mathbb{P}^*(Y(0) = 0 \vee Y(1) = 1 \mid X) \\
&\geq 1 - \max\{1, \mathbb{P}^*(Y(0) = 0 \mid X) + \mathbb{P}^*(Y(1) = 1 \mid X)\} \\
&= \min\{0, \mu(X, 0) - \mu(X, 1)\}.
\end{aligned}
$$

Similarly,

$$
\mathbb{P}^*(Y(0) = 0, Y(1) = 1 \mid X) \geq \min\{0, \mu(X, 1) - \mu(X, 0)\}.
$$

Therefore,

$$
\begin{aligned}
\kappa(X) &\geq \min\{0, \pi_1(X)(1 - \pi_0(X))(\mu(X, 0) - \mu(X, 1)) + \pi_0(X)(1 - \pi_1(X))(\mu(X, 1) - \mu(X, 0))\} \\
&= (\pi_0(X) - \pi_1(X))\tau_-(X),
\end{aligned}
$$

whence $\mathbb{E}[\kappa(X)] \geq \mathbb{E}[(\pi_0(X) - \pi_1(X))\tau_-(X)] = \mathrm{FNA}^-_{\pi_0 \to \pi_1}$, as desired.

We next show that $\mathrm{FNA}^-_{\pi_0 \to \pi_1} \in \mathcal{S}(\mathrm{FNA}_{\pi_0 \to \pi_1}; \mathbb{P})$ by exhibiting a $\mathbb{P}^*$ that recovers it and is compatible with $\mathbb{P}$. First, we let $\mathbb{P}^*$ have the same $X$-distribution as $\mathbb{P}$. Next, for each $X$, if $\pi_1(X) = 1$, we set

$$
\begin{aligned}
\mathbb{P}^*(Y(0) = 1, Y(1) = 0 \mid X) &= \min\{0, \mu(X, 0) - \mu(X, 1)\}, \\
\mathbb{P}^*(Y(0) = 1, Y(1) = 1 \mid X) &= \max\{\mu(X, 0), \mu(X, 1)\}, \\
\mathbb{P}^*(Y(0) = 0, Y(1) = 1 \mid X) &= \min\{0, \mu(X, 1) - \mu(X, 0)\}, \\
\mathbb{P}^*(Y(0) = 0, Y(1) = 0 \mid X) &= 1 - \min\{\mu(X, 1), \mu(X, 0)\},
\end{aligned}
$$

and if $\pi_1(X) = 0$, we set

$$\mathbb{P}^*(Y(0) = 0, Y(1) = 1 \mid X) = \min\{0, \mu(X,1) - \mu(X,0)\},$$
$$\mathbb{P}^*(Y(0) = 0, Y(1) = 0 \mid X) = \max\{\mu(X,0), \mu(X,1)\},$$
$$\mathbb{P}^*(Y(0) = 1, Y(1) = 0 \mid X) = \min\{0, \mu(X,0) - \mu(X,1)\},$$
$$\mathbb{P}^*(Y(0) = 1, Y(1) = 1 \mid X) = 1 - \min\{\mu(X,1), \mu(X,0)\}.$$

Note that in each case, the 4 numbers are nonnegative and always sum to 1, and are therefore form a valid distribution on $\{0,1\}^2$. Moreover, in each case, we have that $\mathbb{P}^*(Y(1) = 1 \mid X) = \mu(X,1)$ and $\mathbb{P}^*(Y(0) \mid X) = \mu(X,0)$. Finally, we set $\mathbb{P}^*(A = 1 \mid X, Y(0), Y(1)) = \mathbb{P}(A = 1 \mid X)$, which ensures that we satisfy unconfoundedness and that $\mu(X,A) = \mathbb{E}[Y \mid X, A]$. Therefore, since the $(X, A)$-distribution as well as all $(Y \mid X, A)$-distributions match, we must have that $\mathbb{P}^*$ is compatible with $\mathbb{P}$. Finally, we note that, under this distribution, we exactly have

$$\kappa(X) = \pi_1(X)(1 - \pi_0(X))\min\{0, \mu(X,0) - \mu(X,1)\} + \pi_0(X)\min\{0, \mu(X,1) - \mu(X,0)\}.$$

Therefore, $\mathrm{FNA}^-_{\pi_0 \to \pi_1} = \mathbb{E}[\kappa(X)] \in \mathcal{S}(\mathrm{FNA}_{\pi_0 \to \pi_1}; \mathbb{P})$.

Next, we show that $\mathrm{FNA}^+_{\pi_0 \to \pi_1} \geq \sup(\mathcal{S}(\mathrm{FNA}_{\pi_0 \to \pi_1}; \mathbb{P}))$. Consider any feasible $\mathbb{P}^*$. Note that

$$\mathbb{P}^*(Y(0) = 1, Y(1) = 0 \mid X) \leq \min\{\mathbb{P}^*(Y(0) = 1 \mid X), \, \mathbb{P}^*(Y(1) = 0 \mid X)\}$$
$$= \min\{\mu(X,0), \, 1 - \mu(X,1)\}.$$

Similarly,

$$\mathbb{P}^*(Y(0) = 0, Y(1) = 1 \mid X) \leq \min\{\mu(X,1), \, 1 - \mu(X,0)\}.$$

Therefore,

$$\kappa(X) \leq \min\{\pi_1(X)(1 - \pi_0(X))\mu(X,0) + \pi_0(X)(1 - \pi_1(X))(1 - \mu(X,0)),$$
$$\pi_1(X)(1 - \pi_0(X))(1 - \mu(X,1)) + \pi_0(X)(1 - \pi_1(X))\mu(X,1)\},$$

the expectation of which is defined to be $\mathrm{FNA}^+_{\pi_0 \to \pi_1}$. Therefore, $\mathbb{E}[\kappa(X)] \leq \mathrm{FNA}^+_{\pi_0 \to \pi_1}$, as desired.

We next show that $\mathrm{FNA}^+_{\pi_0 \to \pi_1} \in \mathcal{S}(\mathrm{FNA}_{\pi_0 \to \pi_1}; \mathbb{P})$ by exhibiting a $\mathbb{P}^*$ that recovers it and is compatible with $\mathbb{P}$. First, we let $\mathbb{P}^*$ have the same $(X, A)$-distribution as $\mathbb{P}$. Next, for each $X$, if $\pi_1(X) = 1$, we set

$$\mathbb{P}^*(Y(0) = 1, Y(1) = 0 \mid X) = \min\{\mu(X,0), \, 1 - \mu(X,1)\},$$
$$\mathbb{P}^*(Y(0) = 1, Y(1) = 1 \mid X) = \max\{0, \, \mu(X,0) + \mu(X,1) - 1\},$$
$$\mathbb{P}^*(Y(0) = 0, Y(1) = 1 \mid X) = \min\{\mu(X,1), \, 1 - \mu(X,0)\},$$
$$\mathbb{P}^*(Y(0) = 0, Y(1) = 0 \mid X) = \max\{0, \, 1 - \mu(X,0) - \mu(X,1)\},$$

and if $\pi_1(X) = 0$, we set

$$\mathbb{P}^*(Y(0) = 0, Y(1) = 0 \mid X) = \min\{\mu(X,1), \, 1 - \mu(X,0)\},$$
$$\mathbb{P}^*(Y(0) = 0, Y(1) = 0 \mid X) = \max\{0, \, \mu(X,0) + \mu(X,1) - 1\},$$
$$\mathbb{P}^*(Y(0) = 1, Y(1) = 0 \mid X) = \min\{\mu(X,0), \, 1 - \mu(X,1)\},$$
$$\mathbb{P}^*(Y(0) = 1, Y(1) = 1 \mid X) = \max\{0, \, 1 - \mu(X,0) - \mu(X,1)\},$$

Note that in each case, the 4 numbers are nonnegative and always sum to 1, and are therefore form a valid distribution on $\{0,1\}^2$. Moreover, in each case, we have that $\mathbb{P}^*(Y(1) = 1 \mid X) = \mu(X,1)$ and $\mathbb{P}^*(Y(0) \mid X) = \mu(X,0)$. Finally, we set $\mathbb{P}^*(A = 1 \mid X, Y(0), Y(1)) = \mathbb{P}(A = 1 \mid X)$, which ensures that we satisfy unconfoundedness and that $\mu(X,A) = \mathbb{E}[Y \mid X, A]$. Therefore, since the $(X, A)$-distribution as well as all $(Y \mid X, A)$-distributions match, we must have that $\mathbb{P}^*$ is compatible with $\mathbb{P}$. Finally, we note that, under this distribution, we exactly have

$$\kappa(X) = \min\{\pi_1(X)(1 - \pi_0(X))\mu(X,0) + \pi_0(X)(1 - \pi_1(X))(1 - \mu(X,0)),$$
$$\pi_1(X)(1 - \pi_0(X))(1 - \mu(X,1)) + \pi_0(X)(1 - \pi_1(X))\mu(X,1)\}.$$

Therefore, $\mathrm{FNA}^+_{\pi_0 \to \pi_1} = \mathbb{E}[\kappa(X)] \in \mathcal{S}(\mathrm{FNA}_{\pi_0 \to \pi_1}; \mathbb{P})$.

To complete the proof, note that $\mathrm{FNA}_{\pi_0 \to \pi_1}$ is linear in $\mathbb{P}^*$ and that $\{\mathbb{P}^* : \mathbb{P}^* \circ \mathcal{C}^{-1} = \mathbb{P}\}$ is a convex set, so that $\mathcal{S}(\mathrm{FNA}_{\pi_0 \to \pi_1}; \mathbb{P})$ is a convex set. $\qquad\square$

## A.3 Proof of Thm. 3

We first present the following restatement of theorems 1 and 2 of [39].

**Lemma 4.** *Let $\mathbb{P}(U), \mathbb{P}(V)$ denote two given distributions on scalar variables. Then,*

$$\sup_{\mathbb{P}^*(U,V):\mathbb{P}^*(U)=\mathbb{P}(U),\,\mathbb{P}^*(V)=\mathbb{P}(V)} \mathbb{P}^*(U-V<\delta) = 1 + \inf_y(\mathbb{P}(U<y+\delta) - \mathbb{P}(V\le y)),$$

$$\inf_{\mathbb{P}^*(U,V):\mathbb{P}^*(U)=\mathbb{P}(U),\,\mathbb{P}^*(V)=\mathbb{P}(V)} \mathbb{P}^*(U-V<\delta) = \sup_y(\mathbb{P}(U<y+\delta) - \mathbb{P}(V\le y)).$$

*Proof.* We start with the restatement of theorems 1 and 2 of [39] given by the right- and left-hand sides of theorem 3.1 of [20], respectively:

$$\inf_{\mathbb{P}^*(U,V):\mathbb{P}^*(U)=\mathbb{P}(U),\,\mathbb{P}^*(V)=\mathbb{P}(V)} \mathbb{P}^*(U-V<\delta) = \sup_y(\mathbb{P}(U<y+\delta) - \mathbb{P}(-V<-y) - 1) \wedge 0,$$

$$\sup_{\mathbb{P}^*(U,V):\mathbb{P}^*(U)=\mathbb{P}(U),\,\mathbb{P}^*(V)=\mathbb{P}(V)} \mathbb{P}^*(U-V<\delta) = \inf_y(\mathbb{P}(U<y+\delta) - \mathbb{P}(-V<-y)) \wedge 1.$$

Then, substituting $\mathbb{P}(-V < -y) = 1 - \mathbb{P}(V \le y)$ and using the fact that $\lim_{y\to-\infty}(\mathbb{P}(U<y+\delta) - \mathbb{P}(V\le y)) = 0$, we obtain the statements above. □

We now turn to proving Thm. 3.

*Proof.* Set

$$\mathcal{M}(\mathbb{P}) = \{\mathbb{P}^*(X,A,Y^*(0),Y^*(1)) : \mathbb{P}^* \circ \mathcal{C}^{-1} = \mathbb{P},\ \mathbb{P}^*(A=1\mid X) = \mathbb{P}^*(A=1\mid X,Y^*(a)),\ a\in\{0,1\}\},$$
$$\mathcal{M}_{Y^*(1),Y^*(0)|X}(\mathbb{P}) = \{\mathbb{P}^*(Y^*(1),Y^*(0)) : \mathbb{P}^*(Y^*(a)\le y) = \mathbb{P}(Y\le y, A=a),\ y\in\mathbb{R}, a\in\{0,1\}\}.$$

Note that

$$\mathcal{M}(\mathbb{P}) = \{\mathbb{P}(X)\times\mathbb{P}(A)\times\mathbb{P}^*(Y^*(0),Y^*(1)\mid X) : \mathbb{P}^*(Y^*(0),Y^*(1)\mid X) \in \mathcal{M}_{Y^*(1),Y^*(0)|X}(\mathbb{P})\}.$$

First, write

$$\sup(\mathcal{S}(\psi_{\zeta,\delta};\mathbb{P})) = \sup_{\mathbb{P}^*\in\mathcal{M}(\mathbb{P})} \mathbb{E}\mathbb{P}^*(\zeta(X)\text{ITE}<\delta\mid X)$$
$$= \mathbb{E}\sup_{\mathbb{P}^*\in\mathcal{M}_{Y^*(1),Y^*(0)|X}(\mathbb{P})} \mathbb{P}^*(\zeta(X)Y^*(1) - \zeta(X)Y^*(0)<\delta\mid X),$$

and similarly for $\inf$. We now consider the inside of the expectation for every $X$ as the sum of two variables $U+V$, where $U = \zeta(X)Y^*(1)$ and $V = -\zeta(X)Y^*(0)$, conditioned on $X$. Then the result follows by Lemma 4. □

## A.4 Proof of Lemma 1

*Proof.* Because ITE $\in \{-1,0,1\}$, we have

$$\text{CVaR}_\alpha(\text{ITE}) = \sup_\beta \big(\beta + \alpha^{-1}\mathbb{P}^*(\text{ITE}=-1)\min\{-1-\beta,\,0\}$$
$$+ \alpha^{-1}\mathbb{P}^*(\text{ITE}=0)\min\{-\beta,\,0\}$$
$$+ \alpha^{-1}\mathbb{P}^*(\text{ITE}=1)\min\{1-\beta,\,0\}\big).$$

Since $\alpha \in (0,1)$, the objective approaches $-\infty$ as $\beta \to \infty$ or $\beta \to -\infty$. Thus, there are only three possible solutions that realize the supremum: $\beta \in \{-1,0,1\}$. Plugging these in above, we obtain

$$\text{CVaR}_\alpha(\text{ITE}) = \max\{-1,\ -\alpha^{-1}\mathbb{P}^*(\text{ITE}=-1),\ 1 - 2\alpha^{-1}\mathbb{P}^*(\text{ITE}=-1) - \alpha^{-1}\mathbb{P}^*(\text{ITE}=0)\}.$$

First, we note that $\mathbb{P}^*(\text{ITE}=-1) = \text{FNA}_{0\to1}$. Second, we note that

$$\mathbb{P}^*(\text{ITE}=0) = \mathbb{P}^*(Y^*(0) = Y^*(1) = 0) + \mathbb{P}^*(Y^*(0) = Y^*(1) = 1)$$
$$= (\mathbb{P}^*(Y^*(1) = 0) - \mathbb{P}^*(Y^*(0) = 1, Y^*(1) = 0))$$
$$+ (\mathbb{P}^*(Y^*(0) = 1) - \mathbb{P}^*(Y^*(0) = 1, Y^*(1) = 0))$$
$$= (1 - \mathbb{E}^*[Y^*(1)] - \text{FNA}_{0\to1}) + (\mathbb{E}^*[Y^*(0)] - \text{FNA}_{0\to1})$$
$$= 1 - \text{ATE} - 2\text{FNA}_{0\to1}.$$

Substituting yields the result. □

# B  Proofs for Sec. 5

## B.1  Preliminaries

**Lemma 5.** *Let $f, g : \mathcal{X} \to \mathbb{R}$ be given. Suppose $f$ satisfies a margin with sharpness $\alpha$. Fix $p \geq 1$. Then, for some $c > 0$,*

$$\mathbb{E}[(\mathbb{I}[g(X) \leq 0] - \mathbb{I}[f(X) \leq 0])f(X)] \leq c\|f - g\|_p^{\frac{p(1+\alpha)}{p+\alpha}}, \tag{10}$$

$$\mathbb{E}[(\mathbb{I}[g(X) \leq 0] - \mathbb{I}[f(X) \leq 0])f(X)] \leq c\|f - g\|_\infty^{1+\alpha}, \tag{11}$$

$$\mathbb{P}(\mathbb{I}[g(X) \leq 0] \neq \mathbb{I}[f(X) \leq 0], f(X) \neq 0) \leq c\|f - g\|_p^{\frac{p\alpha}{p+\alpha}}, \tag{12}$$

$$\mathbb{P}(\mathbb{I}[g(X) \leq 0] \neq \mathbb{I}[f(X) \leq 0], f(X) \neq 0) \leq c\|f - g\|_\infty^{\alpha}. \tag{13}$$

*Proof.* Eqs. (11) and (13) are essentially a restatement of lemma 5.1 of [4]. Their statement focuses on conditional probabilities minus 0.5, but the proof remains identical for real-valued functions. Eq. (10) is essentially a similar restatement of lemma 5.2 of [4].

We conclude by proving Eq. (12): for any $t > 0$,

$$\mathbb{P}(\mathbb{I}[g(X) \leq 0] \neq \mathbb{I}[f(X) \leq 0], f(X) \neq 0)$$
$$\leq \mathbb{P}(0 < |f(X)| \leq t) + \mathbb{P}(\mathbb{I}[g(X) \leq 0] \neq \mathbb{I}[f(X) \leq 0], |f(X)| > t)$$
$$\leq (t/t_0)^\alpha + \mathbb{P}(|f(X) - g(X)| > t)$$
$$\leq (t/t_0)^\alpha + \|f - g\|_p^p t^{-p}.$$

Setting $t = \|f - g\|_p^{\frac{p}{p+\alpha}}$ yields the result. $\qquad\square$

**Lemma 6.** *Fix any $\tilde{\mu}, \tilde{e}, \tilde{\eta}_1, \ldots, \tilde{\eta}_m$ with either $\tilde{\mu} = \mu$ or $\tilde{e} = e$. Set $\kappa_\ell = 1$ if $\tilde{\eta}_\ell = \eta_\ell$ and otherwise set $\kappa_\ell = 0$. Then,*

$$\mathbb{E}[\phi_{g_0,\ldots,g_m}^\rho(X, A, Y; \tilde{e}, \tilde{\mu}, \tilde{\eta}_1, \ldots, \tilde{\eta}_m)] = \text{AHE}_{g_0,\ldots,g_m}^\rho \quad \text{if } \kappa_1 = \cdots = \kappa_m = 1,$$
$$\mathbb{E}[\phi_{g_0,\ldots,g_m}^\rho(X, A, Y; \tilde{e}, \tilde{\mu}, \tilde{\eta}_1, \ldots, \tilde{\eta}_m)] \geq \text{AHE}_{g_0,\ldots,g_m}^\rho \quad \text{if } \rho_\ell = + \text{ whenever } \kappa_\ell = 0,$$
$$\mathbb{E}[\phi_{g_0,\ldots,g_m}^\rho(X, A, Y; \tilde{e}, \tilde{\mu}, \tilde{\eta}_1, \ldots, \tilde{\eta}_m)] \leq \text{AHE}_{g_0,\ldots,g_m}^\rho \quad \text{if } \rho_\ell = - \text{ whenever } \kappa_\ell = 0.$$

*Proof.* Because either $\tilde{\mu} = \mu$ or $\tilde{e} = e$, we have that

$$\mathbb{E}[\phi_{g_0,\ldots,g_m}^\rho(X, A, Y; \tilde{e}, \tilde{\mu}, \tilde{\eta}_1, \ldots, \tilde{\eta}_m)] = \mathbb{E}\Bigg[g_0^{(0)}(X)\mu(X, 0) + g_0^{(1)}(X)\mu(X, 1) + g_0^{(2)}(X)$$
$$+ \sum_{\ell=1}^m \rho_\ell \mathbb{I}[\tilde{\eta}_\ell(X) \leq 0]\eta_\ell(X)\Bigg].$$

If $\kappa_1 = \cdots = \kappa_m = 1$, then the first equation in the statement is immediate.

If $\kappa_\ell = 0$, note that

$$(\mathbb{I}[\tilde{\eta}_\ell(X) \leq 0] - \mathbb{I}[\eta_\ell(X) \leq 0])\eta_\ell(X) = \mathbb{I}[(\tilde{\eta}_\ell(X) \leq 0) \text{ XOR } (\eta_\ell(X) \leq 0)]|\eta_\ell(X)| \geq 0.$$

Therefore, if, among all $\ell$ with $\kappa_\ell = 0$, the sign $\rho_\ell$ is the same, then the biases in $\rho_\ell \mathbb{E}[\mathbb{I}[\tilde{\eta}_\ell(X) \leq 0]\eta_\ell(X)]$ all go the same way, establishing the latter two inequalities in the statement. $\qquad\square$

## B.2  Proof of Lemmas 2 and 3

*Proof of Lemma 2.* If $t > t_0$ then clearly $\mathbb{P}(0 < |f(X)| \leq t) \leq 1$. If $t \leq t_0$ then $\mathbb{P}(0 < |f(X)| \leq t) = \mathbb{P}(f(X) \neq 0) - \mathbb{P}(|f(X)| > t) \leq 0$. Finally note that, $\mathbb{I}[t > t_0] \leq (t/t_0)^\infty$. $\qquad\square$

*Proof of Lemma 3.* Let $M$ be the bound on the derivative of the CDF on $(-\epsilon, 0) \cup (0, \epsilon)$. If $t \geq \epsilon$ then clearly $\mathbb{P}(0 < |f(X)| \leq t) \leq 1$. If $t < \epsilon$ then $\mathbb{P}(0 < |f(X)| \leq t) \leq 2Mt$. Thus, $\mathbb{P}(0 < |f(X)| \leq t) \leq t/\min\{(2M)^{-1}, \epsilon\}$. $\qquad\square$

## B.3 Proof of Thm. 4

*Proof.* We first tackle the first inequality to be proven. We will proceed by bounding each of

$$\left|\mathbb{E}[\phi^\rho_{g_0,\ldots,g_m}(X,A,Y;\check{e},\check{\mu},\check{\eta}_1,\ldots,\check{\eta}_m)] - \mathbb{E}[\phi^\rho_{g_0,\ldots,g_m}(X,A,Y;\tilde{e},\tilde{\mu},\check{\eta}_1,\ldots,\check{\eta}_m)]\right|, \tag{14}$$

$$\left|\mathbb{E}[\phi^\rho_{g_0,\ldots,g_m}(X,A,Y;\tilde{e},\tilde{\mu},\check{\eta}_1,\ldots,\check{\eta}_m)] - \mathbb{E}[\phi^\rho_{g_0,\ldots,g_m}(X,A,Y;\tilde{e},\tilde{\mu},\tilde{\eta}_1,\ldots,\tilde{\eta}_m)]\right|. \tag{15}$$

We begin by bounding Eq. (14) considering separately the case that $\tilde{e} = e$ and that $\tilde{\mu} = \mu$. For brevity let us set

$$\check{\zeta}^{(a)}(X) = g_0^{(a)}(X) + \sum_{\ell=1}^m \rho_\ell \mathbb{I}[\check{\eta}_\ell(X) \le 0]g_\ell^{(a)}(X) \in [-m-1,\, m+1], \quad a = 0, 1.$$

In the case that $\tilde{e} = e$, we bound Eq. (14) by bounding each of

$$\left|\mathbb{E}[\phi^\rho_{g_0,\ldots,g_m}(X,A,Y;\check{e},\check{\mu},\check{\eta}_1,\ldots,\check{\eta}_m)] - \mathbb{E}[\phi^\rho_{g_0,\ldots,g_m}(X,A,Y;e,\check{\mu},\check{\eta}_1,\ldots,\check{\eta}_m)]\right|, \tag{16}$$

$$\left|\mathbb{E}[\phi^\rho_{g_0,\ldots,g_m}(X,A,Y;e,\check{\mu},\check{\eta}_1,\ldots,\check{\eta}_m)] - \mathbb{E}[\phi^\rho_{g_0,\ldots,g_m}(X,A,Y;e,\tilde{\mu},\check{\eta}_1,\ldots,\check{\eta}_m)]\right|. \tag{17}$$

Using iterated expectations to first take expectations with respect to $Y$ and then with respect to $A$, we find that Eq. (16) is equal to

$$\left|\mathbb{E}\left[\check{\zeta}^{(0)}(X)\frac{\check{e}(X) - e(X)}{1 - \check{e}(X)}(\mu(X,0) - \check{\mu}(X,0)) + \check{\zeta}^{(1)}(X)\frac{e(X) - \check{e}(X)}{\check{e}(X)}(\mu(X,1) - \check{\mu}(X,1))\right]\right| \tag{18}$$

$$\le \frac{m+1}{\bar{e}}\|e - \check{e}\|_2(\|\mu(\cdot,0) - \check{\mu}(\cdot,0)\|_2 + \|\mu(\cdot,0) - \check{\mu}(\cdot,0)\|_2)$$

$$\le \frac{2(m+1)}{\bar{e}^{3/2}}\|e - \check{e}\|_2\|\mu - \check{\mu}\|_2.$$

Iterating expectations the same way, we find that Eq. (17) is equal to 0.

In the case that $\tilde{\mu} = \mu$, we bound Eq. (14) by bounding each of

$$\left|\mathbb{E}[\phi^\rho_{g_0,\ldots,g_m}(X,A,Y;\check{e},\check{\mu},\check{\eta}_1,\ldots,\check{\eta}_m)] - \mathbb{E}[\phi^\rho_{g_0,\ldots,g_m}(X,A,Y;\check{e},\mu,\check{\eta}_1,\ldots,\check{\eta}_m)]\right|, \tag{19}$$

$$\left|\mathbb{E}[\phi^\rho_{g_0,\ldots,g_m}(X,A,Y;e,\check{\mu},\check{\eta}_1,\ldots,\check{\eta}_m)] - \mathbb{E}[\phi^\rho_{g_0,\ldots,g_m}(X,A,Y;\tilde{e},\mu,\check{\eta}_1,\ldots,\check{\eta}_m)]\right|. \tag{20}$$

Using iterated expectations to first take expectations with respect to $Y$ and then with respect to $A$, we find that Eq. (19) is again exactly equal to Eq. (18) and the same bound applies. Again, iterating expectations the same way, we find that Eq. (20) is equal to 0.

We now turn to Eq. (15). Using iterated expectations to first take expectations with respect to $Y$ and then with respect to $A$, we find that Eq. (15) is equal to

$$\left|\mathbb{E}\left[\sum_{\ell=1}^m \rho_\ell(\mathbb{I}[\check{\eta}_\ell(X) \le 0] - \mathbb{I}[\tilde{\eta}_\ell(X) \le 0])\eta_\ell(X)\right]\right|$$

$$\le \sum_{\ell=1}^m \mathbb{E}[|(\mathbb{I}[\check{\eta}_\ell(X) \le 0] - \mathbb{I}[\tilde{\eta}_\ell(X) \le 0])\eta_\ell(X)|].$$

We proceed to bound each summand by applying one of Eqs. (10) to (13) of Lemma 5. Consider the $\ell$th term. Suppose $\kappa_\ell = 1$ (*i.e.*, $\tilde{\eta}_\ell = \eta_\ell$). Then applying Eq. (10) if $p < \infty$ and Eq. (11) if $p = \infty$ yields the desired bound. Suppose $\kappa_\ell = 0$. Since $\eta_\ell(X) \in [-3,3]$, we can bound the $\ell$th term by $3\mathbb{P}(\mathbb{I}[\check{\eta}_\ell(X) \le 0] \ne \mathbb{I}[\tilde{\eta}_\ell(X) \le 0]) = 3\mathbb{P}(\mathbb{I}[\check{\eta}_\ell(X) \le 0] \ne \mathbb{I}[\tilde{\eta}_\ell(X) \le 0], \tilde{\eta}_\ell(X) \ne 0)$, where in the last equality we used $\mathbb{P}(\tilde{\eta}_\ell(X) = 0, \check{\eta}_\ell(X) \ne 0) = 0$. Applying Eq. (12) if $p < \infty$ and Eq. (13) if $p = \infty$ yields the desired bound.

We now turn to proving Eq. (15). We proceed by bounding each of the following:

$$\left\|\phi^\rho_{g_0,\ldots,g_m}(X,A,Y;\check{e},\check{\mu},\check{\eta}_1,\ldots,\check{\eta}_m) - \phi^\rho_{g_0,\ldots,g_m}(X,A,Y;\tilde{e},\check{\mu},\check{\eta}_1,\ldots,\check{\eta}_m)\right\|_2, \tag{21}$$

$$\left\|\phi^\rho_{g_0,\ldots,g_m}(X,A,Y;\check{e},\check{\mu},\check{\eta}_1,\ldots,\check{\eta}_m) - \phi^\rho_{g_0,\ldots,g_m}(X,A,Y;\tilde{e},\tilde{\mu},\check{\eta}_1,\ldots,\check{\eta}_m)\right\|_2, \tag{22}$$

$$\left\|\phi^\rho_{g_0,\ldots,g_m}(X,A,Y;\check{e},\check{\mu},\check{\eta}_1,\ldots,\check{\eta}_m) - \phi^\rho_{g_0,\ldots,g_m}(X,A,Y;\tilde{e},\tilde{\mu},\tilde{\eta}_1,\ldots,\tilde{\eta}_m)\right\|_2. \tag{23}$$

Firstly, Eq. (21) is equal to

$$\left\|\check{\zeta}^{(0)}(X)(1-A)\left(\frac{1}{1-\check{e}}-\frac{1}{1-\tilde{e}}\right)(Y-\check{\mu}(X,0))+\check{\zeta}^{(1)}(X)A\left(\frac{1}{\check{e}}-\frac{1}{\tilde{e}}\right)(Y-\check{\mu}(X,1))\right\|_2$$

$$\leq \frac{2(m+1)}{\bar{e}^2}\|\check{e}-\tilde{e}\|_2.$$

Secondly, Eq. (22) is equal to

$$\left\|\check{\zeta}^{(0)}(X)\frac{A-\tilde{e}(X)}{1-\tilde{e}(X)}(\check{\mu}(X,0)-\tilde{\mu}(X,0))+\check{\zeta}^{(1)}(X)\frac{\tilde{e}(X)-A}{\tilde{e}(X)}(\check{\mu}(X,1)-\tilde{\mu}(X,1))\right\|_2$$

$$\leq \frac{m+1}{\bar{e}}(\|\check{\mu}(\cdot,0)-\tilde{\mu}(\cdot,0)\|_2+\|\check{\mu}(\cdot,0)-\tilde{\mu}(\cdot,0)\|_2)\leq \frac{2(m+1)}{\bar{e}^{3/2}}\|\check{\mu}-\tilde{\mu}\|_2.$$

Lastly, Eq. (23) is equal to

$$\left\|\sum_{\ell=1}^m \rho_\ell(\mathbb{I}[\check{\tilde{\eta}}_\ell(X)\leq 0]-\mathbb{I}[\tilde{\eta}_\ell(X)\leq 0])\left(g_\ell^{(0)}(X)\frac{(A-\check{e}(X))\check{\mu}(X,0)+(1-A)Y}{1-\check{e}(X)}\right.\right.$$

$$\left.\left.+g_\ell^{(1)}(X)\frac{(\check{e}(X)-A)\check{\mu}(X,1)+AY}{\check{e}(X)}+g_\ell^{(2)}(X)\right)\right\|_2$$

$$\leq (4\bar{e}^{-1}+1)\sum_{\ell=1}^m \mathbb{P}(\mathbb{I}[\check{\tilde{\eta}}_\ell(X)\leq 0]\neq \mathbb{I}[\tilde{\eta}_\ell(X)\leq 0])^{1/2}$$

$$= (4\bar{e}^{-1}+1)\sum_{\ell=1}^m \mathbb{P}(\mathbb{I}[\check{\tilde{\eta}}_\ell(X)\leq 0]\neq \mathbb{I}[\tilde{\eta}_\ell(X)\leq 0], \tilde{\eta}_\ell(X)\neq 0)^{1/2}.$$

where in the last equality we used $\mathbb{P}(\tilde{\eta}_\ell(X)=0, \check{\tilde{\eta}}_\ell(X)\neq 0)=0$. Applying Lemma 5, using Eq. (12) if $p<\infty$ and Eq. (13) if $p=\infty$, yields the desired bound. $\square$

### B.4 Proof of Thm. 5

*Proof.* For brevity, let $\phi = \phi_{g_0,\ldots,g_m}^\rho$. Define $\mathcal{I}_k = \{i \equiv k-1 \pmod K\}$, $\mathcal{I}_{-k} = \{i \not\equiv k-1 \pmod K\}$, $\hat{\mathbb{E}}_k f(X,A,Y) = \frac{1}{|\mathcal{I}_k|}\sum_{i\in\mathcal{I}_k} f(X_i,A_i,Y_i)$, and $\mathbb{E}_{|-k} f(X,A,Y) = \mathbb{E}[f(X,A,Y)\mid\{(X_i,A_i,Y_i):i\in\mathcal{I}_{-k}\}]$. We then have

$$\hat{\mathbb{E}}_k\phi(X,A,Y;\hat{e}^{(k)},\hat{\mu}^{(k)},\hat{\eta}_1^{(k)},\ldots,\hat{\eta}_m^{(k)})-\hat{\mathbb{E}}_k\phi(X,A,Y;e,\mu,\eta_1^{(k)},\ldots,\eta_m^{(k)})$$

$$= \mathbb{E}_{|-k}\phi(X,A,Y;\hat{e}^{(k)},\hat{\mu}^{(k)},\hat{\eta}_1^{(k)},\ldots,\hat{\eta}_m^{(k)})-\mathbb{E}_{|-k}\phi(X,A,Y;e,\mu,\eta_1^{(k)},\ldots,\eta_m^{(k)}) \qquad (24)$$

$$+ (\hat{\mathbb{E}}_k-\mathbb{E}_{|-k})(\phi(X,A,Y;\hat{e}^{(k)},\hat{\mu}^{(k)},\hat{\eta}_1^{(k)},\ldots,\hat{\eta}_m^{(k)})-\phi(X,A,Y;e,\mu,\eta_1^{(k)},\ldots,\eta_m^{(k)})). \quad (25)$$

We proceed to show that each of Eqs. (24) and (25) are $o_p(1/\sqrt{n})$.

By Thm. 4, we have that Eq. (24) is

$$O_p\left(\left\|e-\hat{e}^{(k)}\right\|_2\left\|\mu-\hat{\mu}^{(k)}\right\|_2+\sum_{\ell=1}^m\left\|\eta_\ell-\hat{\eta}_\ell^{(k)}\right\|_{p_\ell}^{\frac{p\alpha_\ell}{2p+2\alpha_\ell}}\right).$$

So, by our nuisance-estimation assumptions, Eq. (24) is $o_p(1/\sqrt{n})$.

By Chebyshev's inequality conditioned on $\mathcal{I}_{-k}$, we obtain that Eq. (25) is

$$O_p\left(|\mathcal{I}_k|^{-1/2}\left\|\phi(X,A,Y;\hat{e}^{(k)},\hat{\mu}^{(k)},\hat{\eta}_1^{(k)},\ldots,\hat{\eta}_m^{(k)})-\phi(X,A,Y;e,\mu,\eta_1^{(k)},\ldots,\eta_m^{(k)})\right\|_2\right).$$

By our nuisance-estimation assumptions and Thm. 4, we have that $\left\|\phi(X,A,Y;\hat{e}^{(k)},\hat{\mu}^{(k)},\hat{\eta}_1^{(k)},\ldots,\hat{\eta}_m^{(k)})-\phi(X,A,Y;e,\mu,\eta_1^{(k)},\ldots,\eta_m^{(k)})\right\|_2 = o_p(1)$. Thus, Eq. (25) is $o_p(1/\sqrt{n})$.

The first equation is concluded by noting $\frac{1}{n}\sum_{i=1}^{n}\phi(X_i, A_i, Y_i; e, \mu, \eta_1^{(k)}, \ldots, \eta_m^{(k)})) = \frac{1}{K}\sum_{k=1}^{K}\frac{|\mathcal{I}_k|}{n/K}\hat{\mathbb{E}}_k\phi(X, A, Y; e, \mu, \eta_1^{(k)}, \ldots, \eta_m^{(k)}))$.

For the second equation, first note that the first equation together with the central limit theorem imply

$$\sqrt{n}\left(\widehat{\text{AHE}}_{g_0,\ldots,g_m}^{\rho} - \text{AHE}_{g_0,\ldots,g_m}^{\rho}\right) \rightsquigarrow \mathcal{N}(0, \sigma^2), \ \sigma^2 = \text{Var}(\phi(X, A, Y; e, \mu, \eta_1, \ldots, \eta_m)).$$

Therefore, the result is concluded if we can show that $\sqrt{n}\hat{\text{se}} \rightarrow_p \sigma$. Note that $(n-1)\hat{\text{se}}^2 = \frac{1}{n}\sum_{i=1}^{n}\phi_i^2 - (\widehat{\text{AHE}}_{g_0,\ldots,g_m}^{\rho})^2$ and that $\sigma^2 = \mathbb{E}[\phi^2(X, A, Y; e, \mu, \eta_1, \ldots, \eta_m)] - (\text{AHE}_{g_0,\ldots,g_m}^{\rho})^2$. We have already shown that $\widehat{\text{AHE}}_{g_0,\ldots,g_m}^{\rho} \rightarrow_p \text{AHE}_{g_0,\ldots,g_m}^{\rho}$ and continuous mapping implies the same holds for their squares. Next we study the convergence of $\frac{1}{n}\sum_{i=1}^{n}\phi_i^2$. Using $x^2 - y^2 = (x+y)(x-y)$, we bound

$$\left|\hat{\mathbb{E}}_k\phi^2(X, A, Y; \hat{e}^{(k)}, \hat{\mu}^{(k)}, \hat{\eta}_1^{(k)}, \ldots, \hat{\eta}_m^{(k)}) - \hat{\mathbb{E}}_k\phi^2(X, A, Y; e, \mu, \eta_1^{(k)}, \ldots, \eta_m^{(k)})\right|$$

$$\leq \frac{5(m+1)}{\bar{e}}\left|\hat{\mathbb{E}}_k\phi(X, A, Y; \hat{e}^{(k)}, \hat{\mu}^{(k)}, \hat{\eta}_1^{(k)}, \ldots, \hat{\eta}_m^{(k)}) - \hat{\mathbb{E}}_k\phi(X, A, Y; e, \mu, \eta_1^{(k)}, \ldots, \eta_m^{(k)})\right|.$$

Then, following the very same arguments used to prove the first equation we can show that $\frac{1}{n}\sum_{i=1}^{n}\phi_i^2 \rightarrow_p \mathbb{E}[\phi^2(X, A, Y; e, \mu, \eta_1, \ldots, \eta_m)]$. $\square$

## B.5  Proof of Thm. 6

*Proof.* Define

$$\Psi_0 = \mathbb{E}\left[g_0^{(0)}(X)\mu(X, 0) + g_0^{(1)}(X)\mu(X, 1) + g_0^{(2)}(X)\right]$$

$$\Psi_\ell = \mathbb{E}\left[\min\{0, g_\ell^{(0)}(X)\mu(X, 0) + g_\ell^{(1)}(X)\mu(X, 1) + g_\ell^{(2)}(X)\}\right], \quad \ell = 1, \ldots, m.$$

Since $\text{AHE}_{g_0,\ldots,g_m}^{\rho} = \Psi_0 + \sum_{\ell=1}^{m}\rho_\ell\Psi_\ell$, if the efficient influence functions of each of $\Psi_0, \ldots, \Psi_m$ exist and are given by $\psi_0, \ldots, \psi_m$, respectively, then the efficient influence function of $\text{AHE}_{g_0,\ldots,g_m}^{\rho}$ is given by $\psi_0 + \sum_{\ell=1}^{m}\rho_\ell\psi_\ell$.

Fix $\ell = 1, \ldots, m$ and let us derive the efficient influence function of $\Psi_\ell$. Let $\lambda_x(S)$ be a measure on $\mathcal{X}$ dominating $\mathbb{P}(X \in S)$. Let $\lambda_a$ be the counting measure on $\{0, 1\}$. Let $\lambda_y$ be the counting measure on $\{0, 1\}$. Let $\lambda$ be the product measure. Consider the nonparametric model $\mathcal{P}$ consisting of all distributions on $(X, A, Y)$ that are absolutely continuous with respect to $\lambda$. By theorem 4.5 of Tsiatis [57], the tangent space with respect to this model is given by

$$\mathcal{T}_x + \mathcal{T}_a + \mathcal{T}_y,$$
$$\text{where } \mathcal{T}_x = \{f(X) : \mathbb{E}f(X) = 0, \ \mathbb{E}f^2(X) < \infty\},$$
$$\mathcal{T}_a = \{f(X, A) : \mathbb{E}[f(X, A) \mid X] = 0, \ \mathbb{E}f^2(X, A) < \infty\},$$
$$\mathcal{T}_y = \{f(X, A, Y) : \mathbb{E}[f(X, A, Y) \mid X, A] = 0, \ \mathbb{E}f^2(X, A, Y) < \infty\}.$$

Consider any submodel $\mathbb{P}_t \in \mathcal{P}$ passing through $\mathbb{P}_0 = \mathbb{P}$ with density $f_t(x, a, y) = f_t(x)f_t(a \mid x)f_t(y \mid a, x)$ and score $s_t(x, a, y) = \frac{\partial}{\partial t}\log f_t(x, a, y) = s_t(x) + s_t(a \mid x) + s_y(y \mid a, x) = \frac{\partial}{\partial t}\log f_t(x) + \frac{\partial}{\partial t}\log f_t(a \mid x) + \frac{\partial}{\partial t}\log f_t(y \mid x, a)$ belonging to the tangent space $\mathcal{T}$. The efficient influence function is the unique function $\psi_\ell(x, a, y) \in \mathcal{T}$, should it exist, such that

$$\frac{\partial}{\partial t}\Psi_\ell(\mathbb{P}_t)\Big|_{t=0} = \mathbb{E}[\psi_\ell(x, a, y)s_0(x, a, y)]$$

for any such submodel.

Define $\dot{\mu}_t(x, a) = \frac{\partial}{\partial t}\int_y yf_t(y \mid x, a)d\lambda_y(y)$. Then, by product rule, we have

$$\frac{\partial}{\partial t}\Psi_\ell(\mathbb{P}_t)\Big|_{t=0} = \int_x \mathbb{I}[\eta_\ell(x) \leq 0]\left(g_\ell^{(0)}(x)\dot{\mu}_0(x, 0) + g_\ell^{(1)}(x)\dot{\mu}_0(x, 1)\right)f_0(x)d\lambda_x(x)$$

$$+ \int_x \min\{0, \eta_\ell(x)\}s_0(x)f_0(x)d\lambda_x(x),$$

(26)

where we used the fact that $\eta_\ell(x) = 0$ implies $g_\ell^{(0)}(x) = g_\ell^{(1)}(x) = 0$ for almost every $x$.

Note that

$$\dot{\mu}_0(x,1) = \int_a \int_y \frac{a}{e(x)} y s_0(y \mid x,a) f_0(y \mid x,a) f_0(a \mid x) d\lambda_y(y) d\lambda_a(a),$$

$$\dot{\mu}_0(x,0) = \int_a \int_y \frac{1-a}{1-e(x)} y s_0(y \mid x,a) f_0(y \mid x,a) f_0(a \mid x) d\lambda_y(y) d\lambda_a(a).$$

Note further that since densities integrate to 1 at all $t$'s, we have that,

$$\int_y s_0(y \mid x,a) f(y \mid x,a) d\lambda_y(y) = 0, \tag{27}$$

$$\int_a s_0(a \mid x) f(a \mid x) d\lambda_a(a) = 0, \tag{28}$$

$$\int_x s_0(x) f(x) d\lambda_x(x) = 0. \tag{29}$$

Subtracting 0 from the right hand-side of Eq. (26) in the form of the left-hand side of Eq. (27) times $\mathbb{I}[\eta_\ell(x) \leq 0]\left(g_\ell^{(0)}(x)\frac{1-a}{1-e(x)}\mu(x,0) + g_\ell^{(1)}(x)\frac{a}{e(x)}\mu(x,1)\right)$ plus the left-hand side of Eq. (29) times $\Psi_\ell$, we find that

$$\frac{\partial}{\partial t}\Psi_\ell(\mathbb{P}_t)\Big|_{t=0} = \int \left(\mathbb{I}[\eta_\ell(x) \leq 0]\left(g_\ell^{(0)}(x)\frac{1-a}{1-e(x)}(y - \mu(x,0)) + g_\ell^{(1)}(x)\frac{a}{e(x)}(y - \mu(x,1))\right)\right.$$
$$+ \mathbb{I}[\eta_\ell(x) \leq 0]\left(g_\ell^{(0)}(x)\mu(x,0) + g_\ell^{(1)}(x)\mu(x,1) + g_\ell^{(2)}(x)\right)$$
$$\left. - \Psi_\ell\right) s_0(x,a,y) f_0(x,a,y) d\lambda(x,a,y).$$

Since

$$\mathbb{I}[\eta_\ell(x) \leq 0]\left(g_\ell^{(0)}(x)\frac{1-a}{1-e(x)}(y - \mu(x,0)) + g_\ell^{(1)}(x)\frac{a}{e(x)}(y - \mu(x,1))\right) \in \mathcal{T}_y,$$

$$\mathbb{I}[\eta_\ell(x) \leq 0]\left(g_\ell^{(0)}(x)\mu(x,0) + g_\ell^{(1)}(x)\mu(x,1) + g_\ell^{(2)}(x)\right) - \Psi_\ell \in \mathcal{T}_x,$$

we conclude that their sum

$$\psi_\ell(x,a,y) = \mathbb{I}[\eta_\ell(x) \leq 0]\left(g_\ell^{(0)}(x)\left(\mu(x,0) + \frac{1-a}{1-e(x)}(y - \mu(x,0))\right)\right.$$
$$\left. + g_\ell^{(1)}(x)\left(\mu(x,1) + \frac{a}{e(x)}(y - \mu(x,1))\right) + g_\ell^{(2)}(x)\right) - \Psi_\ell$$

is the efficient influence function for $\Psi_\ell$.

Since $\Psi_0$ is just a weighted average of potential outcomes like the ATE, following the same arguments as in theorem 1 of Hahn [23] shows that

$$\psi_0(x,a,y) = g_0^{(0)}(x)\left(\mu(x,0) + \frac{1-a}{1-e(x)}(y - \mu(x,0))\right)$$
$$+ g_0^{(1)}(x)\left(\mu(x,1) + \frac{a}{e(x)}(y - \mu(x,1))\right) + g_0^{(2)}(x) - \Psi_0$$

is the influence function for $\Psi_0$.

The sum of $\psi_0, \psi_1, \ldots, \psi_m$ is exactly $\phi^\rho_{g_0,\ldots,g_m}(X,A,Y;e,\mu,\eta_1,\ldots,\eta_m) - \text{AHE}^\rho_{g_0,\ldots,g_m}$, which completes the proof for the case where $e(X)$ is unknown.

For the model with $e(X)$ fixed and known, the tangent space is given by just $\mathcal{T}_x + \mathcal{T}_y$, that is, the tangent space component corresponding to the $(A \mid X)$-model is the subspace $\{0\}$. Since each $\psi_\ell$ only had components in $\mathcal{T}_x$ and $\mathcal{T}_y$, it still remains within this more restricted tangent space and therefore is still the efficient influence function of $\Psi_\ell$. □

### B.6 Proof of Corollary 1

*Proof.* The only statement left to prove is the regularity of the estimator. This follows from Lemma 25.23 of Van der Vaart [58] because Thm. 5 shows the estimator is asymptotically linear with influence function $\phi^{\rho}_{g_0,\ldots,g_m}(X, A, Y; e, \mu, \eta_1, \ldots, \eta_m)$, and Thm. 6 shows that this is the efficient influence function. $\square$

### B.7 Proof of Thm. 7

*Proof.* The proof is the same as that of Thm. 5 but using Lemma 6 to translate the bias in the limit of the estimator, $\mathbb{E}[\phi^{\rho}_{g_0,\ldots,g_m}(X, A, Y; \tilde{e}, \tilde{\mu}, \tilde{\eta}_1, \ldots, \tilde{\eta}_m)]$, relative to the target $\text{AHE}^{\rho}_{g_0,\ldots,g_m}$. $\square$