# OpenReview forum: "What's the Harm? Sharp Bounds on the Fraction Negatively Affected by Treatment"
_NeurIPS.cc/2022/Conference — NeurIPS 2022 Accept_

### Official Review · Reviewer_KwDm · 2022-07-11

**Rating:** 6
**Confidence:** 3
**Soundness:** 4 excellent
**Presentation:** 3 good
**Contribution:** 2 fair

**Summary:**

This paper outlines theory regarding the estimation of the fraction of units negatively affected by a treatment in the binary treatment and outcome regime. It describes infinite and finite sample results and concludes by employing a simulation and case study to illustrate the proposed bounds in action.


**Questions:**

In the figures, there is presumably an issue with the smoothing tools used to make the plots. The confidence intervals don't appear to overlap with the interpolated point estimates line (e.g., see the blue lines in the leftmost panel of Figure 1).


**Limitations:**

I see real ethical limitations other than the point made earlier that there doesn't seem to be a natural/universal baseline for these bounds in the same way as for other quantities of interest, so it is somewhat unclear how the results should be interpreted in actual experiments.


**Strengths And Weaknesses:**

The paper has strengths. The work is mathematically rigorous and explores several theoretical angles. The simulations and case study are fairly clear. The proposed method could be useful for investigators who sought to know the proportion of units negatively affected by the treatment. I like how the paper highlights both positive and negative results (i.e. how identification is not in general possible, although the main intuition for the result is more or less just that potential outcomes are never jointly observed).

There are a few weaknesses. For example, as acknowledged by the author, the results are focused on the binary case, something which limits the potential use of the proposed methodology (especially since, with binarizations of continuous outcomes, information about harm is being removed).

Second, there seem to be some general limitations of the target quantity of interest. For example, suppose, for 100 units, Y_i(0) \sim N_i(0,1) and Y_i(1) \sim N(0,1). There here is no treatment effect, but on average Y_i(0) would be greater than Y_i(1) 50% of the time (with little uncertainty), so there's a sense in which the FNA can be hard to interpret independently of other information, although in the binary case the situation is easier. Still, there is the question of what is a reasonable baseline for the FNA bounds. For treatment effects, 0 effect is a natural baseline, but for FNA, it seems to be a more difficult question (in the binary case but also much more in the continuous case, so, it seems to me that such bounds would be most useful for cases where averages could be very misleading (such as in the case of continuous, fat-tailed data) in conjunction with marginal effect information.

Finally, there

---

> ### Author Response · Authors · 2022-08-02
> **Response to Reviewer KwDm**
>
> ## Strengths And Weaknesses:
>
> Thank you for recognizing the strengths, rigor, and extensiveness of our paper. We worked hard to make the presentation fair and clear and are glad the results of this were appreciated.
>
> * "as acknowledged by the author, the results are focused on the binary case"
>
> Thank you for the great question. Short answer: we think binary is most practically relevant and helps keep the presentation focused and clear, but we can extend to the general setting! We propose some limited changes in the main text to keep it readable and some further details in the appendix.
>
> Long answer: see another reply below (due to character limit).
>
> * "Second, there seem to be some general limitations of the target quantity of interest."
>
> Thank you for another engaging question. This touches upon a fundamental and almost philosophical question regarding the meaning of potential outcomes and differences between the sharp and weak null. While not our focus, we think this is worth discussing a bit more and providing some relevant references. To first offer a short answer: we think that to the same extent a sharp null can be natural baseline for treatment (after all, that is what the Fisher Randomization Test tests), so is 0 a natural baseline for FNA.
>
> Long answer: see another reply below (due to character limit).
>
> * "Finally, there"
>
> It looks like you didn't finish writing your thought here.
>
> ## Questions:
>
> * "In the figures, there is presumably an issue with the smoothing tools used to make the plots. The confidence intervals don't appear to overlap with the interpolated point estimates line (e.g., see the blue lines in the leftmost panel of Figure 1)."
>
> To be clear, in Figure 1, only the solid line (point estimate) should always be inside the shaded region (confidence interval), while the dashed line (true value) may or may not be inside the shaded region (indicating whether the confidence interval correctly covers the truth). When the PDF is opened in chrome, preview, or acrobat, the solid lines do appear to be inside their respective shaded regions everywhere in Figure 1. Please let us know if there is still an issue, or if instead perhaps we need to better clarify what the solid and dashed lines represent in the figure.
>
> ## Limitations:
>
> * "I see real ethical limitations other than the point made earlier that there doesn't seem to be a natural/universal baseline for these bounds in the same way as for other quantities of interest, so it is somewhat unclear how the results should be interpreted in actual experiments."
>
> This is an important point that we will add. We still think, at a theoretical level, that 0 is a baseline, as truly null treatments also have 0 FNA. At the same time, since FNA is fundamentally unknowable, it is usually impossible to rule out a positive FNA (that is, generally $\mathrm{FNA}^+>0$), so it is indeed a practically relevant question to judge how big is big in a given domain, and that is a question with no universal answer. At the same time, from our own judgment (and we will add this), the most relevant quantity is the lower bound: some positive FNA usually can never be ruled out, but having a significantly positive lower bound is a strong impetus for careful consideration of the treatment as it demonstrably has a negative effect on a significant portion of the population (albeit, whether a "significant" is 5% or 20% is also up to interpretation) and a robust findings like this can strongly support efforts to actually do something to improve fairness. This is also why double validity is so practically relevant: it says we can very strongly believe that what we estimate is a lower bound, even when we make inevitable estimation errors in learning CATE. So, if we have a significant positive estimate then this is very strong evidence of the existence of harm, and we can even say harm to at least what proportion of the population.

---

> > ### Author Response · Authors · 2022-08-02
> > **Response to Reviewer KwDm, extension beyond binary**
> >
> > * "as acknowledged by the author, the results are focused on the binary case"
> >
> > Thank you for the great question. Short answer: we think binary is most practically relevant and helps keep the presentation focused and clear, but we can extend to the general setting! We propose some limited changes in the main text to keep it readable and some further details in the appendix.
> >
> > Long answer: We actually believe binary outcome is the most common setting in real-world applications of program evaluation and have therefore chosen to focus on it, so as to provide the clearest and most-practically relevant presentation in a short paper with already a lot of results. Motivated by your and your fellow reviewers' questions, nonetheless, we thought hard about it and are able to extend our Theorem 2 to the continuous setting. Defining $\mathrm{FNA}=\mathbb P(\mathrm{ITE}<0)$ in the general setting (continuous or binary or anything) and focusing on $\pi_1=1,\pi_0=0$ for simplicity here in the response, the new general result states that $\mathrm{FNA}^-=\mathbb E[\sup_{y\in[-\infty,\infty]}(\mathbb P(Y<y\mid X,A=1)-\mathbb P(Y\leq y\mid X,A=0))]$ and $\mathrm{FNA}^+=1+\mathbb E[\inf_{y\in[-\infty,\infty]}(\mathbb P(Y<y\mid X,A=1)-\mathbb P(Y\leq y\mid X,A=0))]$. Note the difference between $<$ and $\leq$ in the two terms and that $\pm\infty$ are feasible in the sup and inf. For the binary case, ranging $y$ in $\{-1,0,0.5,1\}$ immediately recovers the result in the current Theorem 2. Estimation and inference can also be similarly extended: essentially, provided appropriate margin and boundedness conditions, the above formulation is going to be Neyman orthogonal to the function $y(X)$ that solves for each $X$ the sup (or, inf) inside the over-$X$ expectation simply because the optimization formulation implies zero derivative at the optimum, and we can additionally orthogonalize the counterfactual CDF in the usual doubly-robust way; this $y(X)$ function is essentially the analogue to the function $\mathbb I[\eta_\ell(X)\leq 0]$. However, implementing this for continuous or binary variables is actually different in practice because for binary we _just_ need to learn the conditional mean, so we think it is still best to keep the estimation theory in the main text focused on the binary case, as it is most relevant and is much easier to understand. To sum up, to address your question with revisions: we propose to update Theorem 2 to this new more general form, to provide an extension of the estimation theory to continuous in the appendix, while keeping the estimation theory in the main text focused on binary as it is now.

---

> > ### Author Response · Authors · 2022-08-02
> > **Response to Reviewer KwDm, on interpretation of potential outcomes**
> >
> > * Second, there seem to be some general limitations of the target quantity of interest.
> >
> > Thank you for another engaging question. This touches upon a fundamental and almost philosophical question regarding the meaning of potential outcomes and differences between the sharp and weak null. While not our focus, we think this is worth discussing a bit more and providing some relevant references. To first offer a short answer: we think that to the same extent a sharp null can be natural baseline for treatment (after all, that is what the Fisher Randomization Test tests), so is 0 a natural baseline for FNA.
> >
> > Now the long answer. Potential outcomes postulate that a single unit has two potential values that are co-distributed but never seen together. They can be thought of as random simply in the sense of sampling a random person from a population, and to the extent the person's trajectory into the future is random then we are also sampling this random future, so the population is really of person-futures. This randomness, therefore, captures the intrinsic idiosyncrasies of an individual and their future as it pertains to how treatments affect them. Consider our case study: we are changing the unemployment benefits from 0 to 1 and considering the impact on a binary employment outcome. Consider setting A with $\mathbb P(Y(1)=Y(0)=0)=\mathbb P(Y(1)=Y(0)=1)=1/2$ and setting B with $\mathbb P(Y(1)=1,Y(0)=0)=\mathbb P(Y(1)=0,Y(0)=1)=1/2$ -- these are materially very different settings, despite our not being able to tell them apart (except with some additional information like covariates $X$, which is quite the point of the paper). Setting A says that the change $0\to1$ does not actually do anything to any one person-future (aka, the sharp Fisherian null, $Y(1)=Y(0)$), and the population just contains 50% who will get employed and 50% who will not, regardless of what treatment they get. Setting B says that, while there are overall 50% who will get employed whether we treat everyone with 1 or we treat everyone with 0 (aka, the weak Neymanian null, $\mathbb E[Y(1)]=\mathbb E[Y(0)]$), this actually comes about by completely switching out **which** individuals make up that 50%. (The point of the paper is that by leveraging rich covariates and clever causal machine learning, we can hope to slightly better figure out who exactly those individuals are by looking at this switching within each $X$-value and aggregating, and therefore be better able to distinguish between these two settings, which on the face of it, without further information, are indistinguishable using experimental data alone.) This is all to say that, writing "$(Y_i(0),Y_i(1)) \sim N_i((0,0),I_2)$", as you do (you were not explicit about the correlation between them but later your wrote one is greater than the other 50% of the time so we inferred that you meant independent/uncorrelated normals), should __not__ be understood as "the treatment does nothing because for each outcome we draw the same random variable" but rather should be understood as "the treatment has a very strong effect: for 50% of the population it has a strong positive and for 50% it has a strong negative effect" -- in this setting the treatment actually does something very strong and very meaningful to the individuals. Writing the distribution of the two potential outcomes is simply describing the full population of person-futures. The "baseline" for FNA can in fact be 0, just as the sharp null is the baseline for treatment from the Fisherian perspective -- the permutation test is testing whether we should reject the baseline sharp null hypothesis, not the weak null. If our treatment is that we silently think about the word "zero" or the word "one" in an isolated room then truly this is a treatment of no effect at all with $Y(0)=Y(1)$, i.e., satisfying the sharp null and having an FNA of 0 because it literally does nothing. Similarly, if the treatment is to give someone 100 dollars and then ask what is their wealth immediately after, then $Y(1)=Y(0)+100$ and FNA is still zero. However, if we ask what is their wealth slightly later on, then for all we know, we may have caused some individuals to get addicted to gambling via this cash infusion and they have lost more wealth (even including the 100 dollars) than they would if we did not intervene, so the FNA may be nonzero. It is unlikely but not unfathomable, and our new tools help suss out what is really the case (up to the limits of the data). Some relevant references we will point to are: "A Paradox from Randomization-Based Causal Inference" by Ding, "Randomization Tests for Weak Null Hypotheses in Randomized Experiments" by Wu and Ding, "Beyond the Sharp Null: Permutation Tests Actually Test Heterogeneous Effects" by Caughey, Dafoe, and Miratrix, "Observational Studies" by Rosenbaum, "Randomization Tests" by Edgington and Onghena, and "Permutation Tests" by Good.

---

### Official Review · Reviewer_gJAN · 2022-07-12

**Rating:** 5
**Confidence:** 3
**Soundness:** 3 good
**Presentation:** 2 fair
**Contribution:** 3 good

**Summary:**

This work aims to derive sharp bounds for FNA, a causal quantity measuring the negative impact of switching from one policy to another. Authors show that FNA is not identifiable and therefore derives sharp bounds for different types of FNA. Finally, they perform experiments to show the proposed estimation method can obtain similar performance to ground truth and outperform the straightforward solution.


**Questions:**

1. I wonder what impedes the extension of the proposed method to real-valued outcome.
2. What does function f stands for in the causal inference problem? It is very confusing when I read definition 1.
3. L272, I cannot get why the first equation means double robustness. What does double robustness mean here?

**Limitations:**

Binary outcome and assumptions listed in the paper.

**Strengths And Weaknesses:**

Strengths:

1. This work focuses on an interesting causal parameter, i.e., FNA, which is important in many applications where decision has to be made between two policies.
2. It provides solid theoretical results, showing that FNA is unidentifiable and therefore derives Sharp bounds for it.
3. AHE is introduced to cover all the parameters of interest.

Weakness:

1. This work only considers binary outcome, which can be quite restricted for real-world applications.
2. The mention of CVaR is a surprise since it is not mentioned in Section 2 and seems not quite relevant to the main goal of the paper: estimating FNA.
3. The justification of why the naive method of estimating AHE does not work (L173-180) is not quite understandable from my perspective.
4. The presentation is not quite self-contained. New concepts suddenly appear (e.g., function f in Definition 1) without clear connection to the causal inference problem.
5. Experiments are performed with a limited number of datasets and baselines.

---

> ### Author Response · Authors · 2022-08-02
> **Response to Reviewer gJAN**
>
> ## Strengths:
>
> 1-3. Thank you for recognizing the importance of the problem studied as well the strength of the methodological and theoretical contributions. We strongly believe the work will be of significant interest to the growing causal ML audience in NeurIPS.
>
> ## Weaknesses:
>
> 1. Short answer: we think binary is most practically relevant and helps keep the presentation focused, but we can extend to the general setting! Thank you for the great question. Long answer: see next reply below (due to character limit).
>
> 2. Correct: CVaR is not particularly relevant here, but we thought it is worthwhile to clarify the specific technical relationship to other work on bounding the distribution of individual treatment effects. We appreciate the comment and are open to suggestions to improve readability, but do not think it is really a "weakness". If you think it will improve the readability of the paper, we can certainly move the remark to Section 7 or the appendix.
>
> 3. Thank you for the comment. We can certainly add some more background here to add clarity. This is a standard phenomenon in causal inference with machine learning, and a great example is given in Fig. 1 of Chernozhukov et al. "Double/Debiased Machine Learning ..." (https://arxiv.org/pdf/1608.00060.pdf#page=4). The phenomenon shows up even in simple ATE estimation, where we compare the cross-fitted doubly robust estimator to the "direct method" $\frac1n\sum_{i=1}^n\hat\tau_-(X_i)$ using some estimator $\hat\tau_-$ for CATE -- essentially, the behavior of that estimator directly influences the behavior of the "direct method" whereas the cross-fitted doubly robust estimator is "debiased", i.e., it removes these first-order bias effects. So, what we mean by the text in lines 173-180 is that we too need to find a way to debias our new AHE estimand and that is what we set out to do in that section. Eq (9) is a special debiased formulation of AHE, for which, in Section 5, we establish the special properties of local robustness (Thm 4), efficiency (Thm 5), and double robustness and validity (Thm 6). The extension from ATE-estimation to AHE-estimation is non-trivial, requires deriving a novel debiased formulation, involves unique characterizations like margin condition and its interaction with convergence rates, cannot be simply resolved by knowing propensities due to dependence on $\eta_\ell$, and has new robustness properties like double validity. But the motivation for _why_ to debias is the same.
>
> 4. Thank you for the comment. $f$ in Definition 1 is a stand in for any given generic function of X. We will add "any function" as a qualifier to clarify this. We will also immediately note right after Definition 1 that we will proceed to apply it to the functions $\eta_\ell$ to make the connection to our problem clear. Please let us know if there are other places where we can further clarify what certain symbols are meant to denote.
>
> 5. We respectfully and strongly disagree that the experiments are limited. Not only do we include an illustrative simulation study that clearly verifies the predictions of the theory, we also have a study with a real large-scale dataset. While certainly supervised-learning datasets are bigger, we think what we provide is quite a high bar for causal machine learning papers, where good public data is difficult to obtain. The contribution is largely methodological, and appropriately the key points of Section 6 is to verify the theory, illustrate the benefits of our estimator, and to demonstrate usage in a real dataset. If you think there is an important message that is not currently conveyed experimentally but that another experiment could conceivably demonstrate, please let us know.
>
> ## Questions:
>
> 1. See our response to your Weakness point 1. We can extend and will provide details on this, but keep the focus in the main text on binary, as it is most relevant and keeps the paper readable rather than covering two separate estimation procedures.
>
> 2. See our response to your Weakness point 4. $f$ is any function. Definition 1 is a property of the function (just like if we wrote down a definition for a function $f$ being continuous).  Apologies for the confusion. We proposed some rewording in point 4 above to make this clearer.
>
> 3. Thanks for the question -- this definitely merits some more in-words explanation. Here the double robustness says that even if only one of either $e$ or $\mu$ are consistently estimated, then our AHE estimator is still consistent, provided $\eta_\ell$ are all consistently estimated. It is in fact a so-called partial double robustness because of the need to estimate $\eta_\ell$ consistently; we will qualify it appropriately. And, double validity says that, if $\eta_\ell$ are _not_ consistently estimated, then even if only one of either $e$ or $\mu$ are consistently estimated then our AHE estimator is still consistent for a valid bound. We will add this in-words explanation of the equations after Theorem 6.

---

> > ### Author Response · Authors · 2022-08-02
> > **Response to Reviewer gJAN, Weakness point 1**
> >
> > Response to "weakness point 1":
> >
> > 1. Thank you for the great question. We actually focus on the binary setting as it is in fact the most common in real-world applications of program evaluation, so the specialization to binary offers the most practically relevant and readable results in a short paper with already a lot of results. Motivated by your and your fellow reviewers' questions, nonetheless, we thought hard about it and are able to extend our Theorem 2 to the continuous setting. Defining $\mathrm{FNA}=\mathbb P(\mathrm{ITE}<0)$ in the general setting (continuous or binary or anything) and focusing on $\pi_1=1,\pi_0=0$ for simplicity here in the response, the new general result states that $\mathrm{FNA}^-=\mathbb E[\sup_{y\in[-\infty,\infty]}(\mathbb P(Y<y\mid X,A=1)-\mathbb P(Y\leq y\mid X,A=0))]$ and $\mathrm{FNA}^+=1+\mathbb E[\inf_{y\in[-\infty,\infty]}(\mathbb P(Y<y\mid X,A=1)-\mathbb P(Y\leq y\mid X,A=0))]$. Note the difference between $<$ and $\leq$ in the two terms and that $\pm\infty$ are feasible in the sup and inf. For the binary case, ranging $y$ in $\{-1,0,0.5,1\}$ immediately recovers the result in the current Theorem 2. Estimation and inference can also be similarly extended: essentially, provided appropriate margin and boundedness conditions, the above formulation is going to be Neyman orthogonal to the function $y(X)$ that solves for each $X$ the sup (or, inf) inside the over-$X$ expectation simply because the optimization formulation implies zero derivative at the optimum, and we can additionally orthogonalize the counterfactual CDF in the usual doubly-robust way; this $y(X)$ function is essentially the analogue to the function $\mathbb I[\eta_\ell(X)\leq 0]$. However, implementing this for continuous or binary variables is actually different in practice because for binary we _just_ need to learn the conditional mean, so we think it is still best to keep the estimation theory in the main text focused on the binary case, as it is most relevant and is much easier to understand. To sum up, to address your question with revisions: we propose to update Theorem 2 to this new more general form, to provide an extension of the estimation theory to continuous in the appendix, while keeping the estimation theory in the main text focused on binary as it is now.

---

> > ### Comment · Reviewer_gJAN · 2022-08-09
> > **Acknowledgement**
> >
> > Thanks a lot for your very helpful response. My score will be maintained.

---

> > > ### Author Response · Authors · 2022-08-09
> > > **Response to Reviewer gJAN**
> > >
> > > Thank you for reading the response. We are very glad you found it helpful! To summarize, we think each of the points you list under Weaknesses and Questions are easily addressable. We hope you agree, given the detailed answers to that effect. Thanks again.

---

### Official Review · Reviewer_jNM1 · 2022-07-13

**Rating:** 7
**Confidence:** 1
**Soundness:** 4 excellent
**Presentation:** 2 fair
**Contribution:** 3 good

**Summary:**

I want to state up front that I do not believe I have the appropriate background expertise to give a deep review of this paper and would rely more heavily on other reviewers.

In this paper the authors study how well they can estimate the fraction of a population that has worse outcomes (for binary outcomes) either under two different treatments or two different policies and relying on data from either fully randomized trials or randomization conditioned on observed variables.  They offer theoretical results for when this fraction with negative outcomes is identifiable and bounds on the value, as well as then methods for estimating these bounds from empirical data.  In experiments they show the availablity of real features can improve bounds.


**Questions:**

What makes the seemingly small improvement in estimates in Fig 4 practically significant/impactful?

**Limitations:**

Honestly, as a non-expert, Sections 1-2 were clear and well-written but much of Section 3 onward I doubt would be understandable to researchers not directly working on causality.

**Strengths And Weaknesses:**

S1. I believe this question of how many users are worse off from a particular treatment or change in policy is an important and understudied question, which could be quite impactful, e.g. for reducing churn in new ML trained policies or as alluded to improving fairness amongst underserved slices of users.  While the paper obviously focuses on theoretical contributions in this direction, a clearer articulation of the implications of this work I think would strengthen the paper.

S2. The results seem useful and do not make strong assumptions.

W1. As mentioned below, the paper beyond Section 2 I think will not be accessible by most researchers not working in causality.

W2. The case study results seem to suggest possibly small practical benefit from the use of X.  A discussion of why even small improvements in estimation are valuable would help clarify the impact of the paper.

---

> ### Author Response · Authors · 2022-08-02
> **Response to Reviewer jNM1**
>
> ## Strengths And Weaknesses:
>
> * **S1**. Thank you for recognizing the importance and novelty of the problem we study. Indeed, the work is grounded in rigorous methodological contributions. Nonetheless, we believe one way to more clearly articulate the implications without hype is to explain how practically important our lower bound on FNA is in supporting efforts to advance fairness, as it is irrefutable evidence of harm, and our method very strongly supports reliably lower bounding FNA, both by accounting for aleatoric uncertainty due to the fundamental problem of causal inference and by providing guarantees that we get valid lower bounds even when we make inevitable mistakes in estimation (i.e., double validity in Theorem 6).
>
> * **S2**. Indeed, this is crucial to providing reliable bounds and estimates. Thank you for recognziing the importance of this strength.
>
> * **W1**. Our contributions are indeed grounded in rigor, and, in addition to methods that can be deployed in practice, also contribute intellectually to the causal ML community. We tried our best to nonetheless present the work clearly and concisely while being precise. We are glad that the most important sections -- motivating and setting up the problem -- were clear to you, as they should be to anyone, whether they work in causality or not.
>
> * **W2**. Thank you for the engaging question. In fact, we see a clear and strong benefit to using $X$, but we agree we can certainly better explain it. The most important benefit is seen in Fig. 4(a). In 3 of the settings, not using $X$ leads to a trivial lower bound of 0, while using $X$ gives a meaningful 1-3% lower bound. This means that, without using $X$, we could not detect any clear harm, while using $X$ we can reliably declare that a significant portion are in fact harmed. This is important because it supports efforts to dig deeper to address this harm. Consider the case of pri>pub: the ATE analysis suggests a positive average effect, so one would think this is just a good change. The lower bound analysis implies that, necessarily, there are at least 1-3% of the population that will actually necessarily be harmed by this, which is very different from saying maybe no one is harmed or maybe some are harmed but we don't know. In the other 3 settings, we still see the lower bound roughly doubled when we use $X$, which is actually a big increase. While the upper bounds do not change very materially, they are less actionable, because it is very hard to rule out harm -- it is much more actionable to irrefutably demonstrate harm (lower bounds) and therefore support efforts to address it. See also our response to S1. We will make these points regarding the practical importance of lower bounds clearer.
>
> ## Questions
> "What makes the seemingly small improvement in estimates in Fig 4 practically significant/impactful?"
>
> * See answer to W2.
>
> ## Limitations
> "Honestly, as a non-expert, Sections 1-2 were clear and well-written but much of Section 3 onward I doubt would be understandable to researchers not directly working on causality."
>
> * Thank you -- we worked very hard to make sure the motivation for the paper and the problem setting were clearly communicated, and we are glad to see it paid off. Indeed, Sections 3-5 involve technical results that are important to rigorously support our methods and to engage seriously with NeurIPS's causal ML community. We still nonetheless worked hard to make the whole paper readable and to add in-words explanations. If you point us to places where we can add more, we'd be glad to.

---

> > ### Comment · Reviewer_jNM1 · 2022-08-09
> > **Response**
> >
> > Thank you for your detailed response.  I'll leave my rating as Accept.

---

> > > ### Author Response · Authors · 2022-08-09
> > > **Response to Reviewer jNM1**
> > >
> > > Thank you for reading our response. And, thank you for supporting acceptance. We do hope that the clarifications given in the response offer you more confidence in this rating.

---

### Meta-Review · Area_Chair_tDup · 2022-09-08

**Recommendation:** Accept
**Confidence:** Certain

**Metareview:**

This paper motivates and investigates a novel problem in the context of A/B testing -- specifically, it tries to estimate the fraction of negatively affected individuals beyond average treatment effects. The paper is well-written and does a good job of presenting the technical contributions with sufficient rigour as well as discussing their limitations.

**Award:**

No

---

### Decision · Program_Chairs · 2022-09-14

Accept